# Intriguing Properties of Data Attribution on Diffusion Models

**Xiaosen Zheng**[*1], **Tianyu Pang**[†2], **Chao Du**[†2], **Jing Jiang**[†1], **Min Lin**[2]

[1]Singapore Management University
[2]Sea AI Lab, Singapore

{zhengxs, tianyupang, duchao, linmin}@sea.com; jingjiang@smu.edu.sg

## Abstract

Data attribution seeks to trace model outputs back to training data. With the recent development of diffusion models, data attribution has become a desired module to properly assign valuations for high-quality or copyrighted training samples, ensuring that data contributors are fairly compensated or credited. Several theoretically motivated methods have been proposed to implement data attribution, in an effort to improve the trade-off between computational scalability and effectiveness. In this work, we conduct extensive experiments and ablation studies on attributing diffusion models, specifically focusing on DDPMs trained on CIFAR-10 and CelebA, as well as a Stable Diffusion model LoRA-finetuned on ArtBench. Intriguingly, we report counter-intuitive observations that theoretically *unjustified* design choices for attribution empirically outperform previous baselines by a large margin, in terms of both linear datamodeling score and counterfactual evaluation. Our work presents a significantly more efficient approach for attributing diffusion models, while the unexpected findings suggest that at least in non-convex settings, constructions guided by theoretical assumptions may lead to inferior attribution performance. The code is available at https://github.com/sail-sg/D-TRAK.

## 1 Introduction

Training data plays a pivotal role in determining the behavior of machine learning models. To this end, the goal of *data attribution* is to precisely indicate the importance of each training data in relation to the model outputs of interest. Data attribution has been extensively utilized to interpret model predictions (Koh & Liang, 2017; Yeh et al., 2018; Ilyas et al., 2022), detect poisoning attacks or noisy labels (Hammoudeh & Lowd, 2022a; Lin et al., 2022), curate data (Khanna et al., 2019; Jia et al., 2021; Liu et al., 2021), debug model behavior (Kong et al., 2022), and understand conventional generative models such as GANs and VAEs (Kong & Chaudhuri, 2021; Terashita et al., 2021).

On the other hand, diffusion models have made promising progress on generative tasks (Ho et al., 2020; Song et al., 2021b), and they have gained popularity alongside open-sourced large diffusion models such as Stable Diffusion (Rombach et al., 2022). Numerous applications employ customized variants of Stable Diffusion that are personalized via LoRA (Hu et al., 2022) or ControlNet (Zhang et al., 2023). However, the emerging success and potent ability of diffusion models raise legal and ethical concerns, particularly in domains such as artistic creation where data contributors (e.g., artists) seek fair compensation or credit. In this regard, data attribution acts as an essential module to properly assign valuations for high-quality or copyrighted training samples.

Along the research routine of implementing different methods for data attribution, there is a recurring trade-off between computational scalability and effectiveness (Koh & Liang, 2017; Ghorbani & Zou, 2019; Feldman & Zhang, 2020; Pruthi et al., 2020; Ilyas et al., 2022; Schioppa et al., 2022), especially in non-convex settings. Recently, Park et al. (2023) develop an attribution method called TRAK that is both effective and computationally tractable for large-scale models. Georgiev et al. (2023) makes additional use of TRAK on diffusion models to attribute newly generated images to training data.

In this work, we conduct comprehensive experiments and ablation studies on attributing diffusion models. In addition to taking TRAK as a strong baseline, we evaluate several prevalent attribution

---

[*]Work done during an internship at Sea AI Lab. [†]Corresponding authors.

approaches. Intriguingly, we report counter-intuitive observations that after integrating theoretically *unjustified* design choices into TRAK (the resulting method is named diffusion-TRAK, **D-TRAK**), our D-TRAK method consistently outperforms previous baselines including TRAK, in terms of both linear datamodeling score (Park et al., 2023) and counterfactual evaluation (Hooker et al., 2019; Ilyas et al., 2022). Furthermore, D-TRAK has a number of empirical advantages such as insensitivity to checkpoint selection and fewer timestep requirements, as described in Section 4.

Although D-TRAK is empirically appealing for attributing diffusion models, it is challenging to provide a satisfactory theoretical explanation for questions such as "why D-TRAK performs better than TRAK?" or "are there better design choices than D-TRAK?". Therefore, the unanticipated results reported in this paper suggest that, at least in non-convex settings, theoretically motivated (under simplified assumptions) constructions are not necessarily superior design choices for practical attribution problems, and that the mechanism of data attribution requires a deeper understanding.

## 2 PRELIMINARIES

This section provides a concise overview of diffusion models, the definition of data attribution, the evaluation metrics associated with it, and advanced methods for attribution.

### 2.1 DIFFUSION MODELS

Our research primarily focuses on discrete-time diffusion models, specifically denoising diffusion probabilistic models (DDPMs) (Ho et al., 2020) and latent diffusion models (LDMs) that serve as the foundation of Stable Diffusion (Rombach et al., 2022). Below we briefly recap the notations of DDPMs, where we consider a random variable $\boldsymbol{x} \in \mathcal{X}$ and define a *forward* diffusion process on $\boldsymbol{x}$ as $\boldsymbol{x}_{1:T} \triangleq \boldsymbol{x}_1, \cdots, \boldsymbol{x}_T$ with $T \in \mathbb{N}^+$. The data distribution is $\boldsymbol{x} \sim q(\boldsymbol{x})$ and the Markov transition probability from $\boldsymbol{x}_{t-1}$ to $\boldsymbol{x}_t$ is $q(\boldsymbol{x}_t|\boldsymbol{x}_{t-1}) \triangleq \mathcal{N}(\boldsymbol{x}_t|\sqrt{1-\beta_t}\boldsymbol{x}_{t-1}, \beta_t\mathbf{I})$, where $\boldsymbol{x}_0 = \boldsymbol{x}$ and $\beta_1, \cdots, \beta_T$ correspond to a variance schedule. A notable property of DDPMs is that they can sample $\boldsymbol{x}_t$ at an arbitrary timestep $t$ directly from $\boldsymbol{x}$, since there is $q(\boldsymbol{x}_t|\boldsymbol{x}) = \mathcal{N}(\boldsymbol{x}_t|\sqrt{\overline{\alpha}_t}\boldsymbol{x}, (1-\overline{\alpha}_t)\mathbf{I})$, where $\alpha_t \triangleq 1 - \beta_t$ and $\overline{\alpha}_t \triangleq \prod_{i=1}^t \alpha_i$. Sohl-Dickstein et al. (2015) show that when $\beta_t$ are small, the *reverse* diffusion process can also be modeled by Gaussian conditionals.

Specifically, for the DDPMs framework, the reverse transition probability from $\boldsymbol{x}_t$ to $\boldsymbol{x}_{t-1}$ is written as $p_\theta(\boldsymbol{x}_{t-1}|\boldsymbol{x}_t) = \mathcal{N}(\boldsymbol{x}_{t-1}|\boldsymbol{\mu}_\theta(\boldsymbol{x}_t, t), \sigma_t^2\mathbf{I})$, where $\theta \in \mathbb{R}^d$ is the model parameters and $\sigma_t$ are time dependent constants that can be predefined or analytically computed (Bao et al., 2022). Instead of directly modeling the data prediction $\boldsymbol{\mu}_\theta$, DDPMs choose to model the noise prediction $\boldsymbol{\epsilon}_\theta$ based on the parameterization $\boldsymbol{\mu}_\theta(\boldsymbol{x}_t, t) = \frac{1}{\sqrt{\alpha_t}}\left(\boldsymbol{x}_t - \frac{\beta_t}{\sqrt{1-\overline{\alpha}_t}}\boldsymbol{\epsilon}_\theta(\boldsymbol{x}_t, t)\right)$. The training objective of $\boldsymbol{\epsilon}_\theta(\boldsymbol{x}_t, t)$ can be derived from optimizing the variational bound of negative log-likelihood formulated as follows:

$$\mathcal{L}_{\text{ELBO}}(\boldsymbol{x}; \theta) = \mathbb{E}_{\boldsymbol{\epsilon}, t}\left[\frac{\beta_t^2}{2\sigma_t^2\alpha_t(1-\overline{\alpha}_t)}\left\|\boldsymbol{\epsilon} - \boldsymbol{\epsilon}_\theta(\sqrt{\overline{\alpha}_t}\boldsymbol{x} + \sqrt{1-\overline{\alpha}_t}\boldsymbol{\epsilon}, t)\right\|_2^2\right], \quad (1)$$

where $\boldsymbol{\epsilon} \sim \mathcal{N}(\boldsymbol{\epsilon}|\mathbf{0}, \mathbf{I})$ and $t \sim \mathcal{U}([1, T])$ denotes the discrete uniform distribution between 1 and $T$. Let $\mathcal{D} \triangleq \{\boldsymbol{x}^n\}_{n=1}^N$ be a training dataset that $\boldsymbol{x}^n \sim q(\boldsymbol{x})$, then the empirical training objective on $\mathcal{D}$ can be written as $\mathcal{L}_{\text{ELBO}}(\mathcal{D}; \theta) = \frac{1}{N}\sum_{\boldsymbol{x}^n \in \mathcal{D}} \mathcal{L}_{\text{ELBO}}(\boldsymbol{x}^n, \theta)$. To benefit sample quality, DDPMs apply a simplified training objective that corresponds to a weighted variational bound and is formulated as

$$\mathcal{L}_{\text{Simple}}(\boldsymbol{x}; \theta) = \mathbb{E}_{\boldsymbol{\epsilon}, t}\left[\left\|\boldsymbol{\epsilon} - \boldsymbol{\epsilon}_\theta(\sqrt{\overline{\alpha}_t}\boldsymbol{x} + \sqrt{1-\overline{\alpha}_t}\boldsymbol{\epsilon}, t)\right\|_2^2\right], \quad (2)$$

where the empirical objective on $\mathcal{D}$ is similarly written as $\mathcal{L}_{\text{Simple}}(\mathcal{D}; \theta) = \frac{1}{N}\sum_{\boldsymbol{x}^n \in \mathcal{D}} \mathcal{L}_{\text{Simple}}(\boldsymbol{x}^n, \theta)$.

### 2.2 DATA ATTRIBUTION AND EVALUATION METRICS

Data attribution refers to the goal of tracing model outputs back to training data. We follow Park et al. (2023) and recap the formal definition of data attribution as below:

**Definition 1** (Data attribution). *Consider an ordered training set of samples $\mathcal{D} \triangleq \{\boldsymbol{x}^n\}_{n=1}^N$ and a model output function $\boldsymbol{\mathcal{F}}(\boldsymbol{x}; \theta)$. A data attribution method $\tau(\boldsymbol{x}, \mathcal{D})$ is a function $\tau : \mathcal{X} \times \mathcal{X}^N \to \mathbb{R}^N$ that, for any sample $\boldsymbol{x} \in \mathcal{X}$ and a training set $\mathcal{D}$, assigns a score to each training input $\boldsymbol{x}^n \in \mathcal{D}$ indicating its importance to the model output $\boldsymbol{\mathcal{F}}(\boldsymbol{x}; \theta^*(\mathcal{D}))$, where $\theta^*(\mathcal{D}) = \arg\min_\theta \boldsymbol{\mathcal{L}}(\mathcal{D}; \theta)$.*[1]

---

[1] We apply bold symbols of $\boldsymbol{\mathcal{F}}$ and $\boldsymbol{\mathcal{L}}$ to highlight the model output function and training objective in the *definition of data attribution*, respectively, to distinguish them from the functions used in specific attribution methods.

There have been various metrics to evaluate data attribution methods, including leave-one-out influences (Koh & Liang, 2017; Koh et al., 2019; Basu et al., 2021), Shapley values (Ghorbani & Zou, 2019), and performance on auxiliary tasks (Jia et al., 2021; Hammoudeh & Lowd, 2022a). However, these metrics may pose computational challenges in large-scale scenarios or be influenced by the specific characteristics of the auxiliary task. In light of this, Park et al. (2023) propose the linear datamodeling score (LDS), which considers the sum of attributions as an additive proxy for $\mathcal{F}$, as a new metric for evaluating data attribution methods. In accordance with Definition 1, we define the *attribution-based output prediction* of the model output $\mathcal{F}(\boldsymbol{x}; \theta^*(\mathcal{D}'))$ as

$$g_\tau(\boldsymbol{x}, \mathcal{D}'; \mathcal{D}) \triangleq \sum_{\boldsymbol{x}^n \in \mathcal{D}'} \tau(\boldsymbol{x}, \mathcal{D})_n, \tag{3}$$

where $\mathcal{D}'$ is a subset of $\mathcal{D}$ as $\mathcal{D}' \subset \mathcal{D}$. Then the LDS metric can be constructed as follows:

**Definition 2** (Linear datamodeling score). *Considering a training set $\mathcal{D}$, a model output function $\mathcal{F}(\boldsymbol{x}; \theta)$, and a corresponding data attribution method $\tau$. Let $\{\mathcal{D}^m\}_{m=1}^M$ be $M$ randomly sampled subsets of the training dataset $\mathcal{D}$ that $\mathcal{D}^m \subset \mathcal{D}$, each of size $\alpha \cdot N$ for some $\alpha \in (0, 1)$. The linear datamodeling score (LDS) of a data attribution $\tau$ for a specific sample $\boldsymbol{x} \in \mathcal{X}$ is given by*

$$\mathrm{LDS}(\tau, \boldsymbol{x}) \triangleq \rho\left(\{\mathcal{F}(\boldsymbol{x}; \theta^*(\mathcal{D}^m)) : m \in [M]\}, \{g_\tau(\boldsymbol{x}, \mathcal{D}^m; \mathcal{D}) : m \in [M]\}\right), \tag{4}$$

*where $\rho$ denotes Spearman rank correlation (Spearman, 1987).*

To counter the randomness of the training mechanism (the process of approximating $\theta^*(\mathcal{D}^m)$), for every subset $\mathcal{D}^m$, we train three models with different random seeds and average the model output function. We also consider the counterfactual evaluation to study the utility of different attribution methods, following common practice (Hooker et al., 2019; Feldman & Zhang, 2020; Ilyas et al., 2022; Park et al., 2023; Brophy et al., 2023; Georgiev et al., 2023). We compare the pixel-wise $\ell_2$-distance and CLIP cosine similarity of generated images, using the models trained before/after the exclusion of the highest-ranking positive influencers identified by different attribution methods.

## 2.3 ATTRIBUTION METHODS

The primary interface employed by attribution methods is the score $\tau(\boldsymbol{x}, \mathcal{D})$, which is calculated for each training input to indicate its importance to the output of interest. From robust statistics (Cook & Weisberg, 1982; Hampel et al., 2011), influence functions are a classical concept that approximates how much an infinitesimally up-weighting of a training sample $\boldsymbol{x}^n \in \mathcal{D}$ affects the model output function $\mathcal{F}(\boldsymbol{x}; \theta^*)$, measured on an sample of interest $\boldsymbol{x}$. In the convex setting, Koh & Liang (2017) show that the attributing score of influence function can be computed as $\tau_{\mathrm{IF}}(\boldsymbol{x}, \mathcal{D})_n = \nabla_\theta \mathcal{F}(\boldsymbol{x}; \theta^*)^\top \cdot \mathcal{H}_{\theta^*}^{-1} \cdot \nabla_\theta \mathcal{L}(\boldsymbol{x}^n; \theta^*)$ for $n \in [N]$, where $\mathcal{H}_{\theta^*} = \nabla_\theta^2 \mathcal{L}(\mathcal{D}; \theta^*)$ is the Hessian matrix at the optimal parameters $\theta^*$. Previous work has shown that computing the inverse of the Hessian matrix exhibits numerical instability, particularly when dealing with deep models (Basu et al., 2021; Pruthi et al., 2020). In more recent approaches, the Hessian matrix is substituted with the Fisher information matrix (Ting & Brochu, 2018; Barshan et al., 2020; Teso et al., 2021; Grosse et al., 2023). In contrast, retraining-based methodologies demonstrate greater efficacy in accurately assigning predictions to training data, albeit necessitating the training of numerous models, ranging from thousands to tens of thousands, to achieve desired effectiveness (Ghorbani & Zou, 2019; Feldman & Zhang, 2020; Ilyas et al., 2022).

**Tracing with the randomly-projected after kernel (TRAK).** In a more recent study, Park et al. (2023) develop an approach known as TRAK, which aims to enhance the efficiency and scalability of attributing discriminative classifiers. In the TRAK algorithm, a total of $S$ subsets denoted as $\{\mathcal{D}^s\}_{s=1}^S$ are initially sampled from the training dataset $\mathcal{D}$, where each subset has a fixed size of $\beta \cdot N$ for $\beta \in (0, 1]$.[2] On each subset $\mathcal{D}^s$, a model is trained to obtain the parameters $\theta_s^* \in \mathbb{R}^d$ and a random projection matrix $\mathcal{P}_s$ is sampled from $\mathcal{N}(0, 1)^{d \times k}$ (typically there is $k \ll d$). Then TRAK constructs the projected gradient matrices $\Phi_{\mathrm{TRAK}}^s$ and the weighting terms $\mathcal{Q}_{\mathrm{TRAK}}^s$ as

$$\begin{aligned}
\Phi_{\mathrm{TRAK}}^s &= \left[\phi^s(\boldsymbol{x}^1); \cdots ; \phi^s(\boldsymbol{x}^N)\right]^\top, \text{ where } \phi^s(\boldsymbol{x}) = \mathcal{P}_s^\top \nabla_\theta \mathcal{F}(\boldsymbol{x}; \theta_s^*); \\
\mathcal{Q}_{\mathrm{TRAK}}^s &= \mathrm{diag}\left(Q^s(\boldsymbol{x}^1), \cdots, Q^s(\boldsymbol{x}^N)\right), \text{ where } Q^s(\boldsymbol{x}) = \frac{\partial \mathcal{L}}{\partial \mathcal{F}}(\boldsymbol{x}; \theta_s^*).
\end{aligned} \tag{5}$$

---

[2]The values of $S$ and $\beta$ in TRAK are different from $M$ and $\alpha$ that applied for computing LDS in Definition 2.

Table 1: LDS (%) on CIFAR-2 with different constructions of $\phi^s(\boldsymbol{x})$. All the values of LDS are calculated with $\mathcal{F} = \mathcal{L} = \mathcal{L}_{\text{Simple}}$, and the model is a DDPM with $T = 1000$. We select 10, 100, and 1000 timesteps evenly spaced within the interval $[1, T]$ to approximate the expectation $\mathbb{E}_t$. For each sampled timestep, we sample one standard Gaussian noise $\boldsymbol{\epsilon} \sim \mathcal{N}(\boldsymbol{\epsilon}|\boldsymbol{0}, \mathbf{I})$ to approximate the expectation $\mathbb{E}_{\boldsymbol{\epsilon}}$. The projection dimension of each $\mathcal{P}_s$ is $k = 4096$. While $\phi^s(\boldsymbol{x}) = \mathcal{P}_s^\top \nabla_\theta \mathcal{L}_{\text{Simple}}(\boldsymbol{x}, \theta_s^*)$ should be a reasonable design choice for attributing DDPMs, it is counter-intuitive to observe that using $\phi^s$ constructed by $\mathcal{L}_{\text{Square}}$, $\mathcal{L}_{\text{Avg}}$, $\mathcal{L}_{\text{2-norm}}$, and $\mathcal{L}_{\text{1-norm}}$ consistently achieve higher values of LDS.

| Method | Construction of $\phi^s(\boldsymbol{x})$ | Validation | | | Generation | | |
|---|---|---|---|---|---|---|---|
| | | 10 | 100 | 1000 | 10 | 100 | 1000 |
| TRAK | $\mathcal{P}_s^\top \nabla_\theta \mathcal{L}_{\text{Simple}}(\boldsymbol{x}, \theta_s^*)$ | 10.66 | 19.50 | 22.42 | 5.14 | 12.05 | 15.46 |
| D-TRAK (**Ours**) | $\mathcal{P}_s^\top \nabla_\theta \mathcal{L}_{\text{ELBO}}(\boldsymbol{x}, \theta_s^*)$ | 8.46 | 9.07 | 13.19 | 3.49 | 3.83 | 5.80 |
| | $\mathcal{P}_s^\top \nabla_\theta \mathcal{L}_{\text{Square}}(\boldsymbol{x}, \theta_s^*)$ | 24.78 | 30.81 | 32.37 | 16.20 | 22.62 | 23.94 |
| | $\mathcal{P}_s^\top \nabla_\theta \mathcal{L}_{\text{Avg}}(\boldsymbol{x}, \theta_s^*)$ | 24.91 | 29.15 | 30.39 | 16.76 | 20.82 | 21.48 |
| | $\mathcal{P}_s^\top \nabla_\theta \mathcal{L}_{\text{1-norm}}(\boldsymbol{x}, \theta_s^*)$ | 23.44 | 30.36 | 32.29 | 15.10 | 21.99 | 23.78 |
| | $\mathcal{P}_s^\top \nabla_\theta \mathcal{L}_{\text{2-norm}}(\boldsymbol{x}, \theta_s^*)$ | 24.72 | 30.91 | 32.35 | 15.75 | 22.44 | 23.82 |
| | $\mathcal{P}_s^\top \nabla_\theta \mathcal{L}_{\infty\text{-norm}}(\boldsymbol{x}, \theta_s^*)$ | 5.22 | 11.54 | 22.25 | 3.99 | 8.11 | 15.94 |

Finally, the attribution score $\tau_{\text{TRAK}}(\boldsymbol{x}, \mathcal{D})$ of TRAK is computed by

$$\tau_{\text{TRAK}}(\boldsymbol{x}, \mathcal{D}) = \left[ \frac{1}{S} \sum_{s=1}^{S} \phi^s(\boldsymbol{x})^\top \left( \Phi_{\text{TRAK}}^{s}{}^\top \Phi_{\text{TRAK}}^s \right)^{-1} \Phi_{\text{TRAK}}^{s}{}^\top \right] \left[ \frac{1}{S} \sum_{s=1}^{S} \mathcal{Q}_{\text{TRAK}}^s \right]. \tag{6}$$

In the discriminative classification cases, Park et al. (2023) design the model output function to be $\mathcal{F}(\boldsymbol{x}; \theta) = \log(\exp(\mathcal{L}(\boldsymbol{x}; \theta)) - 1)$ according to the mechanism of logistic regression.

## 3 DATA ATTRIBUTION ON DIFFUSION MODELS

In the context of generative models, such as diffusion models, it is crucial to accurately attribute the generated images to the corresponding training images. This attribution serves the purpose of appropriately assigning credits or valuations, as well as safeguarding copyright (Ghorbani & Zou, 2019; Dai & Gifford, 2023; Wang et al., 2023; Georgiev et al., 2023). Although the study of data attribution has primarily focused on discriminative classification problems, the Definition 1 of data attribution and its evaluation metric LDS in Definition 2 can also be applied to generative cases.

### 3.1 DIFFUSION-TRAK

In our initial trials, we attempt to adopt TRAK for attributing images generated by DDPMs, drawing inspiration from the implementation described by Georgiev et al. (2023).[3] To be specific, given a DDPM trained by minimizing $\mathcal{L}(\mathcal{D}; \theta) = \mathcal{L}_{\text{Simple}}(\mathcal{D}; \theta)$, there is $\theta^*(\mathcal{D}) = \arg\min_\theta \mathcal{L}_{\text{Simple}}(\mathcal{D}; \theta)$ and we set the model output function to be $\mathcal{F}(\boldsymbol{x}; \theta) = \mathcal{L}(\boldsymbol{x}; \theta) = \mathcal{L}_{\text{Simple}}(\boldsymbol{x}, \theta)$. In this particular configuration, when we compute the attribution score of TRAK in Eq. (6), the weighting terms $\mathcal{Q}_{\text{TRAK}}^s = \frac{\partial \mathcal{L}}{\partial \mathcal{F}} = \mathbf{I}$ become identity matrices, and the function $\phi^s$ is constructed as $\phi^s(\boldsymbol{x}) = \mathcal{P}_s^\top \nabla_\theta \mathcal{L}_{\text{Simple}}(\boldsymbol{x}, \theta_s^*)$.

**Counter-intuitive observations.** Intuitively, it seems reasonable to consider TRAK with $\phi^s(\boldsymbol{x}) = \mathcal{P}_s^\top \nabla_\theta \mathcal{L}_{\text{Simple}}(\boldsymbol{x}, \theta_s^*)$ as a suitable approach for attributing DDPMs. This is particularly applicable in the context of our initial trials, where both $\mathcal{L}$ and $\mathcal{F}$ are $\mathcal{L}_{\text{Simple}}$. Nevertheless, we fortuitously observe that *replacing $\phi_s(\boldsymbol{x})$ with alternative functions can result in higher values of LDS*. Specifically, we study a generalization of the TRAK formula, which we refer to as diffusion-TRAK (**D-TRAK**):

$$\tau_{\text{D-TRAK}}(\boldsymbol{x}, \mathcal{D}) = \left[ \frac{1}{S} \sum_{s=1}^{S} \phi^s(\boldsymbol{x})^\top \left( \Phi_{\text{D-TRAK}}^{s}{}^\top \Phi_{\text{D-TRAK}}^s \right)^{-1} \Phi_{\text{D-TRAK}}^{s}{}^\top \right], \tag{7}$$

where $\Phi_{\text{D-TRAK}}^s = \left[ \phi^s(\boldsymbol{x}^1); \cdots ; \phi^s(\boldsymbol{x}^N) \right]^\top$ are the projected gradient matrices. In contrast to the TRAK formula in Eq. (6), D-TRAK allows $\phi^s$ to be constructed from alternative functions, rather than relying on $\mathcal{F}$ as $\phi^s(\boldsymbol{x}) = \mathcal{P}_s^\top \nabla_\theta \mathcal{F}(\boldsymbol{x}; \theta_s^*)$. The weighting terms are eliminated (i.e., $\mathcal{Q}_{\text{D-TRAK}}^s = \mathbf{I}$)

---

[3]Georgiev et al. (2023) focus on attributing noisy images $\boldsymbol{x}_t$, while we attribute the finally generated image $\boldsymbol{x}$.

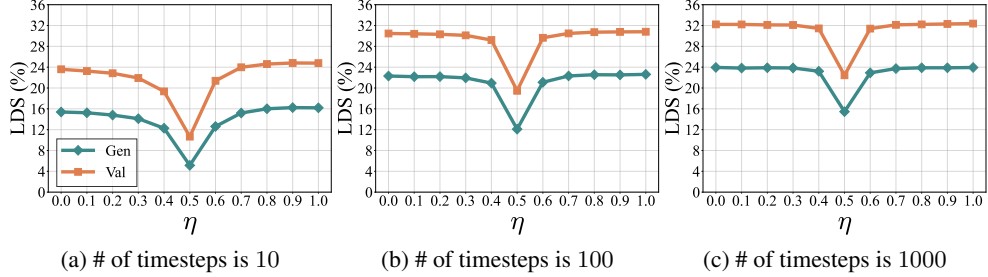

(a) # of timesteps is 10    (b) # of timesteps is 100    (c) # of timesteps is 1000

Figure 1: LDS (%) on CIFAR-2, where $\phi^s$ is constructed by the interpolation described in Section 3.2 for $\eta \in [0, 1]$. The experimental setup employed is identical to that outlined in Table 1. The three subplots are associated with 10, 100, and 1000 timesteps selected to be evenly spaced within the interval $[1, T]$, respectively, which are used to approximate the expectation $\mathbb{E}_t$ over $t \sim \mathcal{U}([1, T])$.

under the assumption that $\mathcal{L}$ and $\mathcal{F}$ are the same. In addition to $\mathcal{L}_{\text{Simple}}$ and $\mathcal{L}_{\text{ELBO}}$, we define $\mathcal{L}_{\text{Square}}$, $\mathcal{L}_{\text{Avg}}$, and $\mathcal{L}_{p\text{-norm}}$ to be the alternative functions constructing $\phi^s$ in D-TRAK, formulated as

$$\mathcal{L}_{\text{Square}}(\boldsymbol{x},\theta)=\mathbb{E}_{t,\boldsymbol{\epsilon}}\Big[\|\boldsymbol{\epsilon}_\theta(\boldsymbol{x}_t,t)\|_2^2\Big]; \ \mathcal{L}_{\text{Avg}}(\boldsymbol{x},\theta)=\mathbb{E}_{t,\boldsymbol{\epsilon}}[\text{Avg}\,(\boldsymbol{\epsilon}_\theta(\boldsymbol{x}_t,t))]; \ \mathcal{L}_{p\text{-norm}}(\boldsymbol{x},\theta)=\mathbb{E}_{t,\boldsymbol{\epsilon}}\Big[\|\boldsymbol{\epsilon}_\theta(\boldsymbol{x}_t,t)\|_p\Big],$$

where $\boldsymbol{x}_t = \sqrt{\overline{\alpha}_t}\boldsymbol{x} + \sqrt{1 - \overline{\alpha}_t}\boldsymbol{\epsilon}$ and $\text{Avg}(\cdot) : \mathcal{X} \to \mathbb{R}$ is the average pooling operation. We instantiate $p = 1, 2, \infty$ for $\mathcal{L}_{p\text{-norm}}$. Our preliminary results are concluded in Table 1, where we train a DDPM with $T = 1000$ on CIFAR-2 (a subset consisting of two classes from CIFAR-10). We compute the values of LDS on the validation set (original test images) and generation set (newly generated images) w.r.t. the training samples. Regarding the trade-off between computational demand and efficiency, we consider different numbers of timesteps (e.g., 10, 100, and 1000) sampled from $t \sim \mathcal{U}([1, T])$ to approximate the expectation of $\mathbb{E}_t$, where these timesteps are selected to be evenly spaced within the interval $[1, T]$ (by the `arange` operation). As can be seen from Table 1, D-TRAK constructed from $\mathcal{L}_{\text{Square}}$, $\mathcal{L}_{\text{Avg}}$, $\mathcal{L}_{2\text{-norm}}$, and $\mathcal{L}_{1\text{-norm}}$ consistently outperform TRAK by a large margin.

## 3.2 INTERPOLATION BETWEEN $\mathcal{L}_{\text{SIMPLE}}$ AND $\mathcal{L}_{\text{SQUARE}}$

The counter-intuitive results in Table 1 pose a challenge in terms of theoretical explanation. It is noteworthy that several alternative functions, namely $\mathcal{L}_{\text{Square}}$, $\mathcal{L}_{\text{Avg}}$, $\mathcal{L}_{2\text{-norm}}$, and $\mathcal{L}_{1\text{-norm}}$, consistently outperform the seemingly reasonable choice of $\mathcal{L}_{\text{Simple}}$. To take a closer look on how these phenomena occur, we take $\mathcal{L}_{\text{Square}}$ as an object of study, and expand the gradients of $\mathcal{L}_{\text{Simple}}$ and $\mathcal{L}_{\text{Square}}$ as

$$\nabla_\theta\mathcal{L}_{\text{Simple}} = \mathbb{E}_{t,\boldsymbol{\epsilon}}\left[2 \cdot (\boldsymbol{\epsilon}_\theta - \boldsymbol{\epsilon})^\top \nabla_\theta\boldsymbol{\epsilon}_\theta\right] \text{ and } \nabla_\theta\mathcal{L}_{\text{Square}} = \mathbb{E}_{t,\boldsymbol{\epsilon}}\left[2 \cdot \boldsymbol{\epsilon}_\theta^\top \nabla_\theta\boldsymbol{\epsilon}_\theta\right], \tag{8}$$

where we omit the dependence on $\boldsymbol{x}$ and $\boldsymbol{\epsilon}$ for the simplicity of notations. We can find that $\nabla_\theta\mathcal{L}_{\text{Simple}}$ and $\nabla_\theta\mathcal{L}_{\text{Square}}$ share the same term of $\nabla_\theta\boldsymbol{\epsilon}_\theta$, and the difference is that they product $\nabla_\theta\boldsymbol{\epsilon}_\theta$ with $2 \cdot (\boldsymbol{\epsilon}_\theta - \boldsymbol{\epsilon})^\top$ and $2 \cdot \boldsymbol{\epsilon}_\theta^\top$, respectively. We deduce that *the information of $\nabla_\theta\boldsymbol{\epsilon}_\theta$ is better retained in the norm-based losses*. Hence, we perform interpolation on these two loss functions and subsequently utilize the resulting function to construct $\phi^s$ in D-TRAK:

$$\phi^s(\boldsymbol{x}) = \mathcal{P}_s^\top\nabla_\theta\left[\eta\mathcal{L}_{\text{Square}}+(1-\eta)\left(\mathcal{L}_{\text{Simple}}-\mathcal{L}_{\text{Square}}\right)\right](\boldsymbol{x},\theta_s^*) = \mathbb{E}_{t,\boldsymbol{\epsilon}}\left[2 \cdot (\eta\boldsymbol{\epsilon}_\theta-(1-\eta)\,\boldsymbol{\epsilon})^\top \nabla_\theta\boldsymbol{\epsilon}_\theta\right],$$

where $\eta \in [0, 1]$ is the interpolation hyperparameter. When $\eta = 0.5$, there is $\phi^s(\boldsymbol{x}) = \frac{1}{2}\mathcal{P}_s^\top\nabla_\theta\mathcal{L}_{\text{Simple}}$ corresponding to TRAK (the constant factor $\frac{1}{2}$ does not affect LDS); when $\eta = 1$, there is $\phi^s(\boldsymbol{x}) = \mathcal{P}_s^\top\nabla_\theta\mathcal{L}_{\text{Square}}$ corresponding to D-TRAK ($\mathcal{L}_{\text{Square}}$) shown in Table 1. Full results of the LDS values w.r.t. various interpolation values $\eta$ are presented in Figure 1. It can be seen that TRAK (i.e., $\eta = 0.5$) has the poorest performance compared to other interpolations. Moreover, as the value of $\eta$ diverges further from $0.5$, approaching either $0$ or $1$, the corresponding LDS values increase.

In Appendix B, we conduct additional ablation studies on the effects of different implementation details. Our findings indicate that at least for the DDPM trained on CIFAR-2, D-TRAK ($\mathcal{L}_{\text{Square}}$) consistently achieves superior performance compared to TRAK. This empirical evidence motivates us to replicate these unexpected findings across various diffusion models and datasets.

## 4 EXPERIMENTS

In this section, we perform comparative analyses between D-TRAK ($\mathcal{L}_{\text{Square}}$) and existing data attribution methods across various settings. The primary metrics employed for evaluating attribution are LDS and counterfactual generations. Additionally, we visualize the attributions for manual inspection. We show that our D-TRAK achieves significantly greater efficacy and computational efficiency.

Table 2: LDS (%) of **retraining-free methods** on CIFAR-2/CIFAR-10 with various # of timesteps (10 or 100). [†] indicates applying TRAK's scalability optimizations as described in Appendix A.3.

| Results on CIFAR-2 | | | | |
|---|---|---|---|---|
| **Method** | Validation | | Generation | |
| | 10 | 100 | 10 | 100 |
| Raw pixel (dot prod.) | $7.77 \pm 0.57$ | | $4.89 \pm 0.58$ | |
| Raw pixel (cosine) | $7.87 \pm 0.57$ | | $5.44 \pm 0.57$ | |
| CLIP similarity (dot prod.) | $6.51 \pm 1.06$ | | $3.00 \pm 0.95$ | |
| CLIP similarity (cosine) | $8.54 \pm 1.01$ | | $4.01 \pm 0.85$ | |
| Gradient (dot prod.) (Charpiat et al., 2019) | $5.14 \pm 0.60$ | $5.07 \pm 0.55$ | $2.80 \pm 0.55$ | $4.03 \pm 0.51$ |
| Gradient (cosine) (Charpiat et al., 2019) | $5.08 \pm 0.59$ | $4.89 \pm 0.50$ | $2.78 \pm 0.54$ | $3.92 \pm 0.49$ |
| TracInCP (Pruthi et al., 2020) | $6.26 \pm 0.84$ | $5.47 \pm 0.87$ | $3.76 \pm 0.61$ | $3.70 \pm 0.66$ |
| GAS (Hammoudeh & Lowd, 2022a) | $5.78 \pm 0.82$ | $5.15 \pm 0.87$ | $3.34 \pm 0.56$ | $3.30 \pm 0.68$ |
| Journey TRAK (Georgiev et al., 2023) | / | / | $7.73 \pm 0.65$ | $12.21 \pm 0.46$ |
| Relative IF[†] (Barshan et al., 2020) | $11.20 \pm 0.51$ | $23.43 \pm 0.46$ | $5.86 \pm 0.48$ | $15.91 \pm 0.39$ |
| Renorm. IF[†] (Hammoudeh & Lowd, 2022a) | $10.89 \pm 0.46$ | $21.46 \pm 0.42$ | $5.69 \pm 0.45$ | $14.65 \pm 0.37$ |
| TRAK (Park et al., 2023) | $11.42 \pm 0.49$ | $23.59 \pm 0.46$ | $5.78 \pm 0.48$ | $15.87 \pm 0.39$ |
| D-TRAK (**Ours**) | **$26.79 \pm 0.33$** | **$33.74 \pm 0.37$** | **$18.82 \pm 0.43$** | **$25.67 \pm 0.40$** |
| Results on CIFAR-10 | | | | |
| **Method** | Validation | | Generation | |
| | 10 | 100 | 10 | 100 |
| Raw pixel (dot prod.) | $2.50 \pm 0.42$ | | $2.25 \pm 0.39$ | |
| Raw pixel (cosine) | $2.71 \pm 0.41$ | | $2.61 \pm 0.38$ | |
| CLIP similarity (dot prod.) | $2.39 \pm 0.41$ | | $1.11 \pm 0.47$ | |
| CLIP similarity (cosine) | $3.39 \pm 0.38$ | | $1.69 \pm 0.49$ | |
| Gradient (dot prod.) (Charpiat et al., 2019) | $0.79 \pm 0.43$ | $0.74 \pm 0.42$ | $1.40 \pm 0.45$ | $1.85 \pm 0.54$ |
| Gradient (cosine) (Charpiat et al., 2019) | $0.66 \pm 0.43$ | $0.58 \pm 0.41$ | $1.24 \pm 0.42$ | $1.82 \pm 0.51$ |
| TracInCP (Pruthi et al., 2020) | $0.98 \pm 0.44$ | $0.96 \pm 0.38$ | $1.26 \pm 0.40$ | $1.39 \pm 0.54$ |
| GAS (Hammoudeh & Lowd, 2022a) | $0.89 \pm 0.48$ | $0.90 \pm 0.38$ | $1.25 \pm 0.41$ | $1.61 \pm 0.54$ |
| Journey TRAK (Georgiev et al., 2023) | / | / | $3.71 \pm 0.37$ | $7.26 \pm 0.43$ |
| Relative IF[†] (Barshan et al., 2020) | $2.76 \pm 0.45$ | $13.56 \pm 0.39$ | $2.42 \pm 0.36$ | $10.65 \pm 0.42$ |
| Renorm. IF[†] (Hammoudeh & Lowd, 2022a) | $2.73 \pm 0.46$ | $12.58 \pm 0.40$ | $2.10 \pm 0.34$ | $9.34 \pm 0.43$ |
| TRAK (Park et al., 2023) | $2.93 \pm 0.46$ | $13.62 \pm 0.38$ | $2.20 \pm 0.38$ | $10.33 \pm 0.42$ |
| D-TRAK (**Ours**) | **$14.69 \pm 0.46$** | **$20.56 \pm 0.42$** | **$11.05 \pm 0.43$** | **$16.11 \pm 0.36$** |

## 4.1 DATASETS

Our experiments are conducted on three datasets including CIFAR ($32 \times 32$), CelebA ($64 \times 64$), and ArtBench ($256 \times 256$). More details of datasets can be found in Appendix A.1.

In addition to the 1,000 held-out validation samples, we created a set of 1,000 generated images for each of the aforementioned datasets. Notably, calculating LDS necessitates retraining a large number of models, which constitutes the majority of the computational cost. In contrast, several attribution methods, such as TRAK and D-TRAK, are computationally efficient and scalable to larger datasets.

## 4.2 BASIC SETUPS OF DIFFUSION MODELS

On CIFAR, we adhere to the original implementation of the unconditional DDPMs (Ho et al., 2020), where the model architecture has 35.7M parameters (i.e., $d = 35.7 \times 10^6$ for $\theta \in \mathbb{R}^d$). The maximum timestep is $T = 1000$, and we choose the linear variance schedule for the forward diffusion process as $\beta_1 = 10^{-4}$ to $\beta_T = 0.02$. We set the dropout rate to $0.1$, employ the AdamW (Loshchilov & Hutter, 2019) optimizer with weight decay of $10^{-6}$, and augment the data with random horizontal flips. A DDPM is trained for 200 epochs with a 128 batch size, using a cosine annealing learning rate schedule with a 0.1 fraction warmup and an initial learning rate of $10^{-4}$. During inference, new images are generated utilizing the 50-step DDIM solver (Song et al., 2021a).

On CelebA, we use an unconditional DDPM implementation that is similar to the one used for CIFAR after being adapted to $64 \times 64$ resolution by slightly modifying the U-Net architecture, which has 118.8M parameters. Other hyper-parameters are identical to those employed on CIFAR.

Table 3: LDS (%) of **retraining-free methods** on ArtBench-2/ArtBench-5 with various # of timesteps (10 or 100). [†] indicates applying TRAK's scalability optimizations as described in Appendix A.3.

| Results on ArtBench-2 | | | | |
|---|---|---|---|---|
| Method | Validation | | Generation | |
| | 10 | 100 | 10 | 100 |
| Raw pixel (dot prod.) | 2.44 ± 0.56 | | 2.60 ± 0.84 | |
| Raw pixel (cosine) | 2.58 ± 0.56 | | 2.71 ± 0.86 | |
| CLIP similarity (dot prod.) | 7.18 ± 0.70 | | 5.33 ± 1.45 | |
| CLIP similarity (cosine) | 8.62 ± 0.70 | | 8.66 ± 1.31 | |
| Gradient (dot prod.) (Charpiat et al., 2019) | 7.68 ± 0.43 | 16.00 ± 0.51 | 4.07 ± 1.07 | 10.23 ± 1.08 |
| Gradient (cosine) (Charpiat et al., 2019) | 7.72 ± 0.42 | 16.04 ± 0.49 | 4.50 ± 0.97 | 10.71 ± 1.07 |
| TracInCP (Pruthi et al., 2020) | 9.69 ± 0.49 | 17.83 ± 0.58 | 6.36 ± 0.93 | 13.85 ± 1.01 |
| GAS (Hammoudeh & Lowd, 2022a) | 9.65 ± 0.46 | 18.04 ± 0.62 | 6.74 ± 0.82 | 14.27 ± 0.97 |
| Journey TRAK (Georgiev et al., 2023) | / | / | 5.96 ± 0.97 | 11.41 ± 1.02 |
| Relative IF[†] (Barshan et al., 2020) | 12.22 ± 0.43 | 27.25 ± 0.34 | 7.62 ± 0.57 | 19.78 ± 0.69 |
| Renorm. IF[†] (Hammoudeh & Lowd, 2022a) | 11.90 ± 0.43 | 26.49 ± 0.34 | 7.83 ± 0.64 | 19.86 ± 0.71 |
| TRAK (Park et al., 2023) | 12.26 ± 0.42 | 27.28 ± 0.34 | 7.78 ± 0.59 | 20.02 ± 0.69 |
| D-TRAK (**Ours**) | **27.61 ± 0.49** | **32.38 ± 0.41** | **24.16 ± 0.67** | **26.53 ± 0.64** |
| Results on ArtBench-5 | | | | |
| Method | Validation | | Generation | |
| | 10 | 100 | 10 | 100 |
| Raw pixel (dot prod.) | 1.84 ± 0.42 | | 2.77 ± 0.80 | |
| Raw pixel (cosine) | 1.97 ± 0.41 | | 3.22 ± 0.78 | |
| CLIP similarity (dot prod.) | 5.29 ± 0.45 | | 4.47 ± 1.09 | |
| CLIP similarity (cosine) | 6.57 ± 0.44 | | 6.63 ± 1.14 | |
| Gradient (dot prod.) (Charpiat et al., 2019) | 4.77 ± 0.36 | 10.02 ± 0.45 | 3.89 ± 0.88 | 8.17 ± 1.02 |
| Gradient (cosine) (Charpiat et al., 2019) | 4.96 ± 0.35 | 9.85 ± 0.44 | 4.14 ± 0.86 | 8.18 ± 1.01 |
| TracInCP (Pruthi et al., 2020) | 5.33 ± 0.37 | 10.87 ± 0.47 | 4.34 ± 0.84 | 9.02 ± 1.04 |
| GAS (Hammoudeh & Lowd, 2022a) | 5.52 ± 0.38 | 10.71 ± 0.48 | 4.48 ± 0.83 | 9.13 ± 1.01 |
| Journey TRAK (Georgiev et al., 2023) | / | / | 7.59 ± 0.78 | 13.31 ± 0.68 |
| Relative IF[†] (Barshan et al., 2020) | 9.77 ± 0.34 | 20.97 ± 0.41 | 8.89 ± 0.59 | 19.56 ± 0.62 |
| Renorm. IF[†] (Hammoudeh & Lowd, 2022a) | 9.57 ± 0.32 | 20.72 ± 0.40 | 8.97 ± 0.58 | 19.38 ± 0.66 |
| TRAK (Park et al., 2023) | 9.79 ± 0.33 | 21.03 ± 0.42 | 8.79 ± 0.59 | 19.54 ± 0.61 |
| D-TRAK (**Ours**) | **22.84 ± 0.37** | **27.46 ± 0.37** | **21.56 ± 0.71** | **23.85 ± 0.74** |

On ArtBench, we fine-tune the Stable Diffusion model (Rombach et al., 2022) using LoRA (Hu et al., 2022) with the rank set to 128, which consists of 25.5M parameters. To fit the resolution ratio of ArtBench, we use a Stable Diffusion checkpoint that has been adapted from $512 \times 512$ to $256 \times 256$ resolution. We train the model in a class-conditional manner where the textual prompt is simply set as "a {class} painting" such as "a romanticism painting". We also set the dropout rate to $0.1$, employ the AdamW optimizer with weight decay of $10^{-6}$, and augment the data with random horizontal flips. We train the model for 100 epochs under a batch size of 64, using a cosine annealing learning rate schedule with $0.1$ fraction warmup and an initial learning rate of $3 \times 10^{-4}$. At the inference phase, we sample new images using the 50-step DDIM solver and a guidance scale of $7.5$ (Ho & Salimans, 2022). More implementation details on training and generation of diffusion models are in Appendix A.1.

## 4.3 EVALUATING LDS FOR ATTRIBUTION METHODS

We set up the LDS benchmarks as described in Appendix A.2. In the following experiments, we report the mean and standard deviation of the LDS scores based on bootstrapping (Johnson, 2001) corresponding to the random re-sampling from the subsets. Regarding the design choices of attribution methods, as shown in Figure 2, for D-TRAK, the best LDS scores are obtained at the final checkpoint. However, for TRAK, using the final checkpoint is not the best choice. Finding which checkpoint yields the best LDS score requires computing the attributions for many times, which means much more computational cost. In practice, we might also only get access to the final model. On ArtBench-2, though TRAK obtains comparable LDS scores at the best checkpoint using 1000 timesteps, we believe that even ignoring the cost of searching the best checkpoint, computing the gradients for 1000 times per sample is unrealistic. In summary, when computing gradients, we use

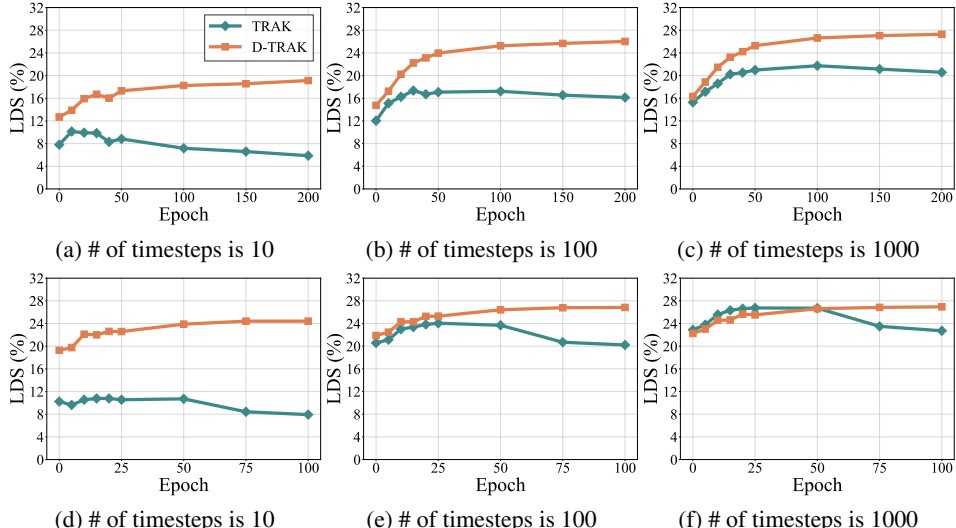

(a) # of timesteps is 10     (b) # of timesteps is 100     (c) # of timesteps is 1000

(d) # of timesteps is 10     (e) # of timesteps is 100     (f) # of timesteps is 1000

Figure 2: The LDS(%) on the generation set of (**Top**) CIFAR-2 and (**Bottom**) Artbench-2 using checkpoints of different epochs. We select 10, 100, and 1000 timesteps evenly spaced within the interval $[1, T]$ to approximate $\mathbb{E}_t$. For each selected timestep, we sample one standard Gaussian noise to approximate $\mathbb{E}_\epsilon$. We set $k = 32768$. For the full results, please check Figures 6, 7 and 8 in Appendix B.

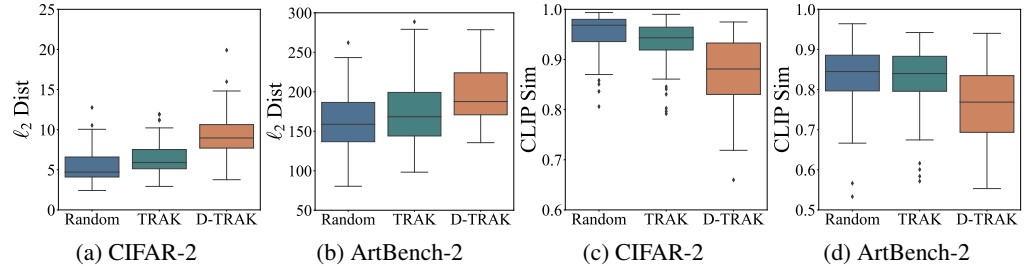

(a) CIFAR-2     (b) ArtBench-2     (c) CIFAR-2     (d) ArtBench-2

Figure 3: Boxplots of counterfactual evaluation on CIFAR-2 and ArtBench-2. We quantify the impact of removing the 1,000 highest-scoring training samples and re-training according to Random, TRAK, and D-TRAK. We measure the pixel-wise $\ell_2$-distance and CLIP cosine similarity between 60 synthesized samples and corresponding images generated by the re-trained models when sampling from the same random seed. For the results on CelebA, please check Figure 11 in Appendix C.3.

the final model checkpoint as default following Koh & Liang (2017); Park et al. (2023); Grosse et al. (2023). We consider using 10 and 100 timesteps selected to be evenly spaced within the interval $[1, T]$. For example, the selected timesteps are $\{1, 101, 201, ..., 901\}$ when the number of timesteps is 10. For each timestep, we sample one standard Gaussian noise. The projection dimension is $k = 32768$.

Following (Hammoudeh & Lowd, 2022b), we consider the attribution baselines that can be applied to our settings. We filter out methods that are intractable on our settings like Leave-One-Out (Cook & Weisberg, 1982) and Shapely Value (Ghorbani & Zou, 2019). We opt out those methods that are inapplicable to DDPMs, which are generally designed for specific tasks and models like Representer Point (Yeh et al., 2018). The baselines could be grouped into retraining-free and retraining-based methods. We further group the retraining-free methods into similarity-based, gradient-based (without kernel) and gradient-based (with kernel) methods. For TRAK and D-TRAK, the retraining-free setting means $S = 1$ and $\beta = 1.0$, where we compute the gradients reusing the model we want to interpret. More implementation details of baselines can be found in Appendix A.3.

As shown in Tables 2, 3, 6, the similarity-based baselines including Raw pixel and CLIP similarity, and those gradient-based (without kernel) methods such as Gradient, TracInCP and GAS, yield poor LDS scores. However, gradient-based (with kernel) methods like TRAK and D-TRAK perform consistently better than other methods. Methods normalizing the attributions based on the magnitude of gradients including Relative Influence and Renormalized Influence have similar performance

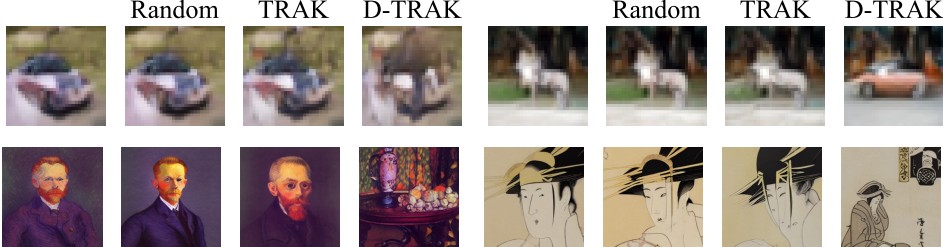

Figure 4: Counterfactual visualization on (**Top**) CIFAR-2 and (**Bottom**) ArtBench-2. We compare the samples to those generated by retrained models using the same seed. See Appendix D.1 for more cases.

compared to TRAK. The Journey TRAK yields low LDS scores as expected because it is designed for attributing noisy images $x_t$ along the sampling trajectory originally, while we attribute the finally generated image $x$. On the validation set, with 10/100 timesteps, D-TRAK achieves improvements of +15.37%/+10.15%, +11.55%/+8.67% and +15.35%/+5.1% on CIFAR-2, CelebA and ArtBench-2, respectively. On the generation set, with 10/100 timesteps, D-TRAK exhibits gains of +13.04%/+9.8%, +9.82%/+6.76%, and +16.38%/+6.51% in the LDS scores for the above three datasets. Regarding CIFAR-10 and ArtBench-5, overall the LDS scores are lower, which is expected because attributing to a larger training set might be more difficult. Nonetheless, D-TRAK still performs significantly better than TRAK. It is also interesting to see that across all the settings, D-TRAK with 10 timesteps computation budget outperforms TRAK with 100 timesteps computation budget. In Tables 7 and 8, we compare D-TRAK with the retraining-based methods including TRAK (ensemble), Empirical Influence and Datamodel, on CIFAR-2, CelebA and ArtBench-2. D-TRAK performs consistently better. More implementation details and empirical results of LDS evaluation can be found in Appendix C.2.

## 4.4 COUNTERFACTUAL EVALUATION

To evaluate the faithfulness of D-TRAK more intuitively, on CIFAR-2, CelebA, and ArtBench-2, we measure the pixel-wise $\ell_2$-dist.[4] and CLIP cosine similarity between images generated by the models trained before/after the exclusion of the top-1000 positive influencers identified by different attribution methods. For both D-TRAK and TRAK, we consider 100 timesteps, sample one standard Gaussian noise, and set $k = 32768$. We consider a control baseline called Random, which randomly removes 1000 training samples before retraining. As shown in Figure 3 and 11, when examining the median pixel-wise $\ell_2$ distance resulting from removing-and-retraining, D-TRAK yields 8.97, 15.07, and 187.61 for CIFAR-2, CelebA, and ArtBench-2, respectively, in contrast to TRAK's values of 5.90, 11.02 and 168.37. D-TRAK obtains median similarities of 0.881, 0.896 and 0.769 for the above three datasets respectively, which are notably lower than TRAK's values of 0.943, 0.942 and 0.840, highlighting the effectiveness of our method. We manually compare the original generated samples to those generated from the same random seed with the re-trained models.[5] As shown in Figure 4, our results suggest our method can better identify images that have a larger impact on the target image.

## 5 DISCUSSION

In this work, we empirically demonstrate that when the model output function of interest $\mathcal{F}$ is the same as the training objective $\mathcal{L}$, *the gradients of $\nabla_\theta \mathcal{L}$ may not be a good design choice for attributing $\mathcal{L}$ itself*. Although this deduction seems counter-intuitive, we find potentially related phenomena observed in previous research. For examples, Park et al. (2023) have reported that using $\nabla_\theta \mathcal{L}$ in TRAK leads to worse performance than using $\nabla_\theta \log(\exp(\mathcal{L}(x; \theta)) - 1)$ on classification problems, even if the LDS score is computed by setting $\mathcal{F} = \mathcal{L}$. However, the TRAK paper has not extensively investigated this observation, and the utilization of the design involving $\nabla_\theta \log(\exp(\mathcal{L}(x; \theta)) - 1)$ is based on theoretical justifications. Furthermore, existing literature on adversarial attacks has also identified the presence of gradient obfuscation when utilizing gradients derived from $\nabla_x \mathcal{L}$ (Athalye et al., 2018). Conversely, employing gradients from alternative functions has been shown to enhance the efficiency of adversarial attacks (Carlini & Wagner, 2017; Croce & Hein, 2020). The aforementioned phenomena have not been adequately explained in a formal manner. We advocate that further investigation be conducted to gain a deeper understanding of these counter-intuitive observations and to enhance the development of data attribution methods.

---

[4]On ArtBench-2, we measure the $\ell_2$-dist. of VQ-VAE encoding for generated images (Rombach et al., 2022).

[5]We also visualize top attributions for sampled target samples in Appendix D.2.

## ACKNOWLEDGMENTS

This research is supported by the Ministry of Education, Singapore, under its Academic Research Fund Tier 2 (Proposal ID: T2EP20222-0047). Any opinions, findings and conclusions or recommendations expressed in this material are those of the authors and do not reflect the views of the Ministry of Education, Singapore.

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

## A    IMPLEMENTATION DETAILS

In this paper, we train various diffusion models for different datasets using the `Diffusers` library.[6] We compute the per-sample gradient following a tutorial of the `PyTorch` library (version 2.0.1).[7] We use the `trak` library[8] to project gradients with a random projection matrix, which is implemented using a faster custom CUDA kernel.[9] For all of our experiments, we use 64 CPU cores and NVIDIA A100 GPUs each with 40GB of memory.

### A.1    DATASETS AND MODELS

**CIFAR (32×32).** The CIFAR-10 dataset (Krizhevsky et al., 2009) contains 50,000 training samples.[10] We randomly sample 1,000 validation samples from CIFAR-10's test set for LDS evaluation. To reduce computation, we also construct a CIFAR-2 dataset as a subset of CIFAR-10, which consists of 5,000 training samples randomly sampled from CIFAR-10's training samples corresponding to the "automobile" and "horse" classes, and 1,000 validation samples randomly sampled from CIFAR-10's test set corresponding to the same two classes.

On CIFAR, we adhere to the original implementation of unconditional DDPMs (Ho et al., 2020), where the model architecture has 35.7M parameters (i.e., $d = 35.7 \times 10^6$ for $\theta \in \mathbb{R}^d$).[11] The maximum timestep is $T = 1000$, and we choose the linear variance schedule for the forward diffusion process as $\beta_1 = 10^{-4}$ to $\beta_T = 0.02$. We set the dropout rate to 0.1, employ the AdamW (Loshchilov & Hutter, 2019) optimizer with weight decay of $10^{-6}$, and augment the data with random horizontal flips. A DDPM is trained for 200 epochs with a 128 batch size, using a cosine annealing learning rate schedule with a 0.1 fraction warmup and an initial learning rate of $10^{-4}$. During inference, new images are generated utilizing the 50-step DDIM solver (Song et al., 2021a).

**CelebA (64×64).** We sample a subset of 5,000 training samples and 1,000 validation samples from the original training set and test set of CelebA (Liu et al., 2015),[12] respectively, and first center crop the images to 140×140 according to Song et al. (2021b) and then resize them to 64×64.

On CelebA, we use an unconditional DDPM implementation that is similar to the one used for CIFAR after being adapted to 64×64 resolution by slightly modifying the U-Net architecture, which has 118.8M parameters.[13] Other hyper-parameters are identical to those employed on CIFAR.

**ArtBench (256×256).** ArtBench (Liao et al., 2022) is a dataset for artwork generation.[14] It includes 60,000 artwork images for 10 distinct artistic styles, with 5,000 training images and 1,000 testing images per style. We construct ArtBench-2 as a subset of ArtBench, consisting of 5,000 training/1,000 validation samples sampled from 10,000 training/2,000 test samples of the "post-impressionism" and "ukiyo-e" classes. We also test our method on ArtBench-5 as a subset of ArtBench, consisting of 12,500 training/1,000 validation samples sampled from 25,000 training/5,000 test samples of the "post-impressionism", "ukiyo-e", "romanticism", "renaissance" and "baroque" classes.

On ArtBench, we fine-tune the Stable Diffusion model (Rombach et al., 2022) using LoRA (Hu et al., 2022) with the rank set to 128, which consists of 25.5M parameters.[15] To fit the resolution ratio of ArtBench, we use a Stable Diffusion checkpoint that has been adapted from 512×512 to 256×256 resolution.[16] We train the model in a class-conditional manner where the textual prompt is simply set as "a {class} painting" such as "a romanticism painting". We also set the dropout rate to 0.1, employ

---

[6] https://github.com/huggingface/diffusers/tree/v0.16.1
[7] https://pytorch.org/tutorials/intermediate/per_sample_grads.html
[8] https://github.com/MadryLab/trak/releases/tag/v0.1.3
[9] https://pypi.org/project/fast-jl/0.1.3
[10] https://huggingface.co/datasets/cifar10
[11] https://huggingface.co/google/ddpm-cifar10-32
[12] https://mmlab.ie.cuhk.edu.hk/projects/CelebA.html
[13] https://github.com/huggingface/diffusers/blob/v0.16.1/examples/unconditional_image_generation/train_unconditional.py
[14] https://github.com/liaopeiyuan/artbench
[15] https://github.com/huggingface/diffusers/blob/v0.16.1/examples/text_to_image/train_text_to_image_lora.py
[16] https://huggingface.co/lambdalabs/miniSD-diffusers

the AdamW optimizer with weight decay of $10^{-6}$, and augment the data with random horizontal flips. We train the model for 100 epochs under a batch size of 64, using a cosine annealing learning rate schedule with 0.1 fraction warmup and an initial learning rate of $3 \times 10^{-4}$. At the inference phase, we sample new images using the 50-step DDIM solver and a guidance scale of 7.5 (Ho & Salimans, 2022).

## A.2 LDS EVALUATION SETUP

To conduct the LDS evaluation, we sample $M = 64$ different random subsets of the training set $\mathcal{D}$, and train three models with different random seeds on each one of these subsets. Each subset sampled to 50% of the size of the training set, that is, we set $\alpha = 0.5$. We then compute the linear datamodeling score for each sample of interest as the Spearman rank correlation between the model output and the attribution-based output prediction of the model output as described in Eq. (3). Especially, when computing $\mathcal{L}_{\text{Simple}}(\boldsymbol{x}; \theta)$ as described in Eq. (2) for any sample of interest from either the validation set or generation set, we consider all the 1000 timesteps selected to be evenly spaced within the interval $[1, T]$ to approximate the expectation $\mathbb{E}_t$. For each timestep, we sample three standard Gaussian noises $\boldsymbol{\epsilon} \sim \mathcal{N}(\boldsymbol{\epsilon}|\mathbf{0}, \mathbf{I})$ to approximate the expectation $\mathbb{E}_{\boldsymbol{\epsilon}}$. Finally, we average the LDS across samples of interest from the validation set and generation set respectively.

## A.3 BASELINES

In this paper, we majorly focus on conducting data attribution in a post-hoc manner, which refers to the application of attribution methods after model training. Post-hoc attribution methods do not add extra restrictions to how we train models and are thus preferred in practice (Ribeiro et al., 2016).

Following (Hammoudeh & Lowd, 2022b), we consider the attribution baselines that can be applied to our settings. We filter out intractable methods on our settings like Leave-One-Out (Cook & Weisberg, 1982) and Shapely Value (Shapley et al., 1953; Ghorbani & Zou, 2019). We exclude those methods that are inapplicable to DDPMs, which are generally designed for specific tasks and models like Representer Point (Yeh et al., 2018). We also omit HYDRA (Chen et al., 2021), which is closely related to TracInCP Pruthi et al. (2020) and can be viewed as trading (incremental) speed for lower precision compared to TracInCP as described by Hammoudeh & Lowd (2022b).

We also noticed two concurrent works that do not apply to our settings. Dai & Gifford (2023) propose to conduct training data attribution on diffusion models based on machine unlearning (Bourtoule et al., 2021). However, their method is restricted to the diffusion models that are trained by a specially designed machine unlearning training process thus is not post-hoc and could not be applied to the common settings. Wang et al. (2023) claim that analyzing training influences in a post-hoc manner using existing influence estimation methods is currently intractable and instead propose a method that also requires training/tuning the pretrained text-to-image model in a specifically designed process, which is called "customization" in their paper.

Regarding methods that use the Hessian matrix as the kernel including Influence Functions (Koh & Liang, 2017), Relative Influence (Barshan et al., 2020) and Renormalized Influence (Hammoudeh & Lowd, 2022a), previous work has shown that computing the inverse of the Hessian matrix exhibits numerical instability in practical scenarios, often resulting in either divergence or high computational costs, particularly when dealing with deep models (Basu et al., 2021; Pruthi et al., 2020). In more recent approaches, the Hessian matrix is substituted with the Fisher information matrix (FIM) (Ting & Brochu, 2018; Barshan et al., 2020; Teso et al., 2021; Grosse et al., 2023). Martens (2020) discuss the relationship between Hessian and FIM more thoroughly.

Recently, Park et al. (2023) developed an estimator that leverages a kernel matrix that is similar to the FIM based on linearizing the model and further employs the classical random projection to speed up the hessian-based influence functions (Koh & Liang, 2017), as the estimators are intractable otherwise. We summarize these two modifications as **TRAK's scalability optimizations** and also apply them to Relative Influence and Renormalized Influence.

We divided the baselines into two categories: retraining-free and retraining-based methods. The retraining-free methods are further classified into three types: similarity-based, gradient-based (without kernel) and gradient-based (with kernel) methods. Raw pixel and CLIP similarity (Radford et al., 2021) are two similarity-based methods. Gradient (Charpiat et al., 2019), TracInCP (Pruthi et al., 2020), and GAS (Hammoudeh & Lowd, 2022a) are gradient-based (without kernel) meth-

ods. TRAK (Park et al., 2023), Relative Influence Barshan et al. (2020) and Renormalized Influence(Hammoudeh & Lowd, 2022a), and Journey TRAK (Georgiev et al., 2023) are gradient-based (with kernel) methods. TRAK (ensemble), Empirical Influence (Feldman & Zhang, 2020), and Datamodel (Ilyas et al., 2022) are retraining-based methods.

We next provide definition and implementation details of the baselines used in Section 4.

**Raw pixel.** This is a naive similarity-based attribution method that simply uses the raw image as the representation and then computes the dot product or cosine similarity between the sample of interest and each training sample as the attribution score. Especially, for ArtBench, which uses latent diffusion models (Rombach et al., 2022), we represent the image using the VAE encodings (Van Den Oord et al., 2017) of the raw image.

**CLIP Similarity.** This is another similarity-based attribution method. We encode each sample into an embedding using CLIP (Radford et al., 2021) and then compute the dot product or cosine similarity between the target sample and each training sample as the attribution score.

**Gradient.** This is a gradient-based influence estimator from Charpiat et al. (2019), which computes the dot product or cosine similarity using the gradient representations of the sample of interest and each training sample, as the attribution score as follows

$$\tau(\boldsymbol{x}, \mathcal{D})_n = \mathcal{P}^\top \nabla_\theta \mathcal{L}_{\text{Simple}}(\boldsymbol{x}; \theta^*)^\top \cdot \mathcal{P}^\top \nabla_\theta \mathcal{L}_{\text{Simple}}(\boldsymbol{x}^n; \theta^*);$$
$$\tau(\boldsymbol{x}, \mathcal{D})_n = \frac{\mathcal{P}^\top \nabla_\theta \mathcal{L}_{\text{Simple}}(\boldsymbol{x}; \theta^*)^\top \cdot \mathcal{P}^\top \nabla_\theta \mathcal{L}_{\text{Simple}}(\boldsymbol{x}^n; \theta^*)}{\|\mathcal{P}^\top \nabla_\theta \mathcal{L}_{\text{Simple}}(\boldsymbol{x}; \theta^*)\| \|\mathcal{P}^\top \nabla_\theta \mathcal{L}_{\text{Simple}}(\boldsymbol{x}^n; \theta^*)\|}.$$

**TracInCP.** We use the TracInCP estimator from Pruthi et al. (2020), implemented as

$$\tau(\boldsymbol{x}, \mathcal{D})_n = \frac{1}{C} \Sigma_{c=1}^C \mathcal{P}_c^\top \nabla_\theta \mathcal{L}_{\text{Simple}}(\boldsymbol{x}; \theta^c)^\top \cdot \mathcal{P}_c^\top \nabla_\theta \mathcal{L}_{\text{Simple}}(\boldsymbol{x}^n; \theta^c),$$

where $C$ is the number of model checkpoints selected from the training trajectory evenly and $\theta^c$ is the corresponding checkpoint. We use four checkpoints from the training trajectory. For example, for CIFAR-2, we take the checkpoints at epochs $\{50, 100, 150, 200\}$.

**GAS.** This is a "renormalized" version of the TracInCP based on using the cosine similarity instead of raw dot products (Hammoudeh & Lowd, 2022a).

**TRAK.** As discussed in Section 3, we adapt the TRAK (Park et al., 2023) to the diffusion setting as described in Eq. (6), the retraining-free TRAK is implemented as

$$\Phi_{\text{TRAK}} = \left[ \phi(\boldsymbol{x}^1); \cdots ; \phi(\boldsymbol{x}^N) \right]^\top, \text{ where } \phi(\boldsymbol{x}) = \mathcal{P}^\top \nabla_\theta \mathcal{L}_{\text{Simple}}(\boldsymbol{x}; \theta^*);$$
$$\tau(\boldsymbol{x}, \mathcal{D})_n = \mathcal{P}^\top \nabla_\theta \mathcal{L}_{\text{Simple}}(\boldsymbol{x}; \theta^*)^\top \cdot \left( \Phi_{\text{TRAK}}^\top \Phi_{\text{TRAK}} + \lambda I \right)^{-1} \cdot \mathcal{P}^\top \nabla_\theta \mathcal{L}_{\text{Simple}}(\boldsymbol{x}^n; \theta^*),$$

where the $\lambda I$ serves for numerical stability and regularization effect. We explore the effect of this term in Appendix B. The *ensemble* version of TRAK is implemented as

$$\Phi_{\text{TRAK}}^s = \left[ \phi^s(\boldsymbol{x}^1); \cdots ; \phi^s(\boldsymbol{x}^N) \right]^\top, \text{ where } \phi^s(\boldsymbol{x}) = \mathcal{P}_s^\top \nabla_\theta \mathcal{L}_{\text{Simple}}(\boldsymbol{x}; \theta_s^*);$$
$$\tau(\boldsymbol{x}, \mathcal{D})_n = \frac{1}{S} \Sigma_{s=1}^S \mathcal{P}_s^\top \nabla_\theta \mathcal{L}_{\text{Simple}}(\boldsymbol{x}; \theta_s^*)^\top \cdot \left( \Phi_{\text{TRAK}}^s{}^\top \Phi_{\text{TRAK}}^s + \lambda I \right)^{-1} \cdot \mathcal{P}_s^\top \nabla_\theta \mathcal{L}_{\text{Simple}}(\boldsymbol{x}^n; \theta_s^*).$$

**Relative Influence.** Barshan et al. (2020) propose the $\theta$-relative influence functions estimator, which normalizes Koh & Liang (2017) influence functions' estimator by HVP magnitude. We adapt this method to our setting after applying TRAK's scalability optimizations, which is formulated as

$$\tau(\boldsymbol{x}, \mathcal{D})_n = \frac{\mathcal{P}^\top \nabla_\theta \mathcal{L}_{\text{Simple}}(\boldsymbol{x}; \theta^*)^\top \cdot \left( \Phi_{\text{TRAK}}^\top \Phi_{\text{TRAK}} + \lambda I \right)^{-1} \cdot \mathcal{P}^\top \nabla_\theta \mathcal{L}_{\text{Simple}}(\boldsymbol{x}^n; \theta^*)}{\| \left( \Phi_{\text{TRAK}}^\top \Phi_{\text{TRAK}} + \lambda I \right)^{-1} \cdot \mathcal{P}^\top \nabla_\theta \mathcal{L}_{\text{Simple}}(\boldsymbol{x}^n; \theta^*) \|}.$$

**Renormalized Influence.** Hammoudeh & Lowd (2022a) also propose to renormalize the influence by the magnitude of the training sample's gradients. We adapt this method to our setting after applying

TRAK's scalability optimizations, which is formulated as

$$\tau(\boldsymbol{x}, \mathcal{D})_n = \frac{\mathcal{P}^\top \nabla_\theta \mathcal{L}_{\text{Simple}}(\boldsymbol{x}; \theta^*)^\top \cdot \left(\Phi_{\text{TRAK}}{}^\top \Phi_{\text{TRAK}} + \lambda I\right)^{-1} \cdot \mathcal{P}^\top \nabla_\theta \mathcal{L}_{\text{Simple}}(\boldsymbol{x}^n; \theta^*)}{\|\mathcal{P}^\top \nabla_\theta \mathcal{L}_{\text{Simple}}(\boldsymbol{x}^n; \theta^*)\|}.$$

**Journey TRAK.** Georgiev et al. (2023) focus on attributing noisy images $\boldsymbol{x}_t$, while we attribute the finally generated image $\boldsymbol{x}$. We adapt their method to our setting by averaging the attributions over the generation timesteps as follows

$$\tau(\boldsymbol{x}, \mathcal{D})_n = \frac{1}{T'} \Sigma_{t=1}^{T'} \mathcal{P}^\top \nabla_\theta \mathcal{L}_{\text{Simple}}^t(\boldsymbol{x}_t; \theta^*)^\top \cdot \left(\Phi_{\text{TRAK}}{}^\top \Phi_{\text{TRAK}} + \lambda I\right)^{-1} \cdot \mathcal{P}^\top \nabla_\theta \mathcal{L}_{\text{Simple}}(\boldsymbol{x}^n; \theta^*),$$

where $T'$ is inference steps set as $50$ and $\boldsymbol{x}_t$ is the noisy generated image along the sampling trajectory.

**Empirical Influence.** We use the downsampling-based approximation to leave-one-out influences as used by Feldman & Zhang (2020), using up to $S = 512$ models trained on different random 50% subsets of the full training set, which is a difference-in-means estimator given by:

$$\tau(\boldsymbol{x}, \mathcal{D})_n = \mathbb{E}_{\mathcal{D}^s \ni \boldsymbol{x}^n} \mathcal{L}_{\text{Simple}}(\boldsymbol{x}; \theta_s^*) - \mathbb{E}_{\mathcal{D}^s \not\ni \boldsymbol{x}^n} \mathcal{L}_{\text{Simple}}(\boldsymbol{x}; \theta_s^*).$$

Especially, when computing $\mathcal{L}_{\text{Simple}}(\boldsymbol{x}; \theta)$ as described in Eq. (2) for any sample of interest from either the validation set or generation set, we consider all the 1000 timesteps selected to be evenly spaced within the interval $[1, T]$ to approximate the expectation $\mathbb{E}_t$. For each timestep, we sample one standard Gaussian noise $\epsilon \sim \mathcal{N}(\epsilon | \boldsymbol{0}, \mathbf{I})$ to approximate the expectation $\mathbb{E}_\epsilon$.

**Datamdoel.** We use the regression-based estimator from Ilyas et al. (2022), using up to $S = 512$ mdoels trained on different random 50% subsets of the full training set.

## A.4 TIME MEASUREMENTS

We evaluate the computational cost of attribution methods using two factors: the total number of timesteps used and the total number of trained models used. These factors are hardware and implementation-agnostic. We note the number of timesteps used as $K$. Unlike classification models, we need to compute the gradients of multiple timesteps per sample for diffusion models. For retraining-free methods, the time it takes to compute attribution scores will be dominated by the time of computing the gradients for different timesteps.

`TRAIN_TIME`: The time to train one model (from scratch or fine-tuning).

`GRAD_TIME`: The time to compute gradients of one model for each sample at each timestep in the entire dataset including train, validation, and generation sets.

`PROJ_TIME`: The time to project gradients of one model for each sample in the entire dataset including training, validation, and generation sets. Note that for each sample, we first average the gradient vectors over multiple timesteps and then conduct the random projection.

We approximate the total compute time for each method as follows.

**TracInCP and GAS:** $C \times (K \times$ `GRAD_TIME` $+$ `PROJ_TIME`$)$

**TRAK and D-TRAK:** $K \times$ `GRAD_TIME` $+$ `PROJ_TIME`

**TRAK and D-TRAK (ensemble):** $S \times ($`TRAIN_TIME` $+ K \times$ `GRAD_TIME` $+$ `PROJ_TIME`$)$

**Empirical Influence and Datamodel:** $S \times$ `TRAIN_TIME`

Table 4: LDS (%) on CIFAR2 corresponding to different model output function of interest $\mathcal{F}$ and different $\phi^s$ in D-TRAK. When computing $\phi^s$, we use 100 timesteps selected to be evenly spaced within the interval $[1, T]$, which are used to approximate the expectation $\mathbb{E}_t$. For each sampled timestep, we sample one standard Gaussian noise $\epsilon \sim \mathcal{N}(\epsilon|\mathbf{0}, \mathbf{I})$ to approximate the expectation $\mathbb{E}_\epsilon$. The projection dimension is $k = 4096$. Here we omit the $(\boldsymbol{x}, \theta_s^*)$ when writing the expressions of $\mathcal{F}$ and $\phi^s$.

| $\phi^s$ \ $\mathcal{F}$ | Validation | | | | | Generation | | | | |
|---|---|---|---|---|---|---|---|---|---|---|
| | $\mathcal{L}_{\text{Simple}}$ | $\mathcal{L}_{\text{ELBO}}$ | $\mathcal{L}_{\text{Square}}$ | $\mathcal{L}_{\text{1-norm}}$ | $\mathcal{L}_{\infty\text{-norm}}$ | $\mathcal{L}_{\text{Simple}}$ | $\mathcal{L}_{\text{ELBO}}$ | $\mathcal{L}_{\text{Square}}$ | $\mathcal{L}_{\text{1-norm}}$ | $\mathcal{L}_{\infty\text{-norm}}$ |
| $\mathcal{P}_s^\top \nabla_\theta \mathcal{L}_{\text{Simple}}$ | 19.50 | 20.54 | 0.16 | 0.10 | -0.06 | 12.05 | 11.88 | 0.25 | 0.24 | 0.51 |
| $\mathcal{P}_s^\top \nabla_\theta \mathcal{L}_{\text{ELBO}}$ | 9.07 | 15.65 | 0.13 | 0.11 | -0.13 | 3.83 | 8.35 | 0.06 | -0.05 | -0.04 |
| $\mathcal{P}_s^\top \nabla_\theta \mathcal{L}_{\text{Square}}$ | 30.81 | 27.06 | 0.57 | 0.55 | 0.57 | 22.62 | 18.24 | 0.15 | 0.14 | 0.30 |
| $\mathcal{P}_s^\top \nabla_\theta \mathcal{L}_{\text{1-norm}}$ | 30.36 | 27.40 | 0.55 | 0.59 | 0.28 | 21.99 | 18.46 | 0.54 | 0.50 | 0.58 |
| $\mathcal{P}_s^\top \nabla_\theta \mathcal{L}_{\infty\text{-norm}}$ | 11.54 | 13.53 | -0.23 | -0.21 | -0.49 | 8.11 | 10.08 | -0.45 | -0.49 | -0.36 |

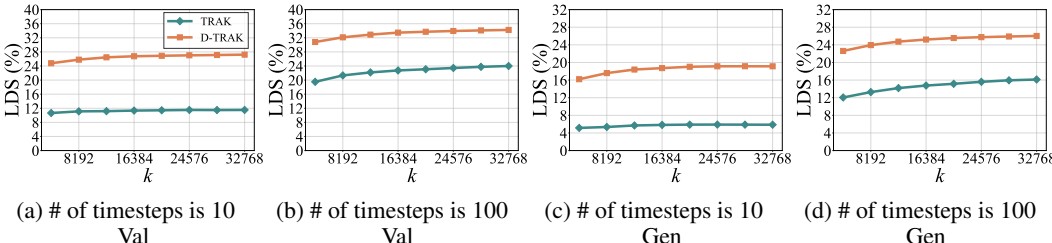

| (a) # of timesteps is 10 Val | (b) # of timesteps is 100 Val | (c) # of timesteps is 10 Gen | (d) # of timesteps is 100 Gen |
|---|---|---|---|

Figure 5: The LDS(%) on CIFAR-2 under different $k$. The number of timesteps is in the parentheses. We consider 10 and 100 timesteps selected to be evenly spaced within the interval $[1, T]$, which are used to approximate the expectation $\mathbb{E}_t$. For each sampled timestep, we sample one standard Gaussian noise $\epsilon \sim \mathcal{N}(\epsilon|\mathbf{0}, \mathbf{I})$ to approximate the expectation $\mathbb{E}_\epsilon$.

# B  ABLATION STUDIES

To double-check if the counter-intuitive observations in Table 1 and Figure 1 are caused by certain implementation details, we perform more ablation studies to compare the performance between TRAK and D-TRAK ($\mathcal{L}_{\text{Square}}$). In this section, we choose the CIFAR-2 as our major setting, we also consider CelebA and ArtBench-2 in some experiments. The details of these settings can be found in Appendix A.1. We set up the corresponding LDS benchmarks as described in Appendix A.2.

We focus on the retraining-free setting, which means $S = 1$ and $\beta = 1.0$, where we compute the gradients reusing the model we want to interpret. When computing gradients for attribution methods, we consider various number of timesteps where the timesteps are selected to be evenly spaced within the interval $[1, T]$ by the `arange` operation, which we term as *uniform*. More concretely, the selected timesteps are $\{1, 101, 201, \cdots, 901\}$ when the number of timesteps is 10. For each timestep, we sample one standard Gaussian noise.

$\mathcal{F}$ **Selection.** Recap the LDS metric in Definition 2, we need to specify the model output function before conducting the LDS evaluation. This introduces a design choice for discussion. Given a DDPM trained by minimizing $\boldsymbol{\mathcal{L}}(\mathcal{D}; \theta) = \mathcal{L}_{\text{Simple}}(\mathcal{D}; \theta)$, there is $\theta^*(\mathcal{D}) = \arg\min_\theta \mathcal{L}_{\text{Simple}}(\mathcal{D}; \theta)$ and setting the model output function to be $\mathcal{F}(\boldsymbol{x}; \theta) = \boldsymbol{\mathcal{L}}(\boldsymbol{x}; \theta) = \mathcal{L}_{\text{Simple}}(\boldsymbol{x}, \theta)$ should be a natural choice. Nevertheless, we further conduct a comprehensive study by constructing different LDS benchmarks that use different model output functions such as $\mathcal{L}_{\text{ELBO}}$, $\mathcal{L}_{\text{Squared}}$, $\mathcal{L}_{\text{1-norm}}$, and $\mathcal{L}_{\infty\text{-norm}}$ and then evaluate different constructions of $\phi^s$ as what we did in Table 1. As shown in Table 4, we first observe that setting the model output function as one of $\mathcal{L}_{\text{Squared}}$, $\mathcal{L}_{\text{1-norm}}$, and $\mathcal{L}_{\infty\text{-norm}}$, will make the attribution task fail as indicated by near-zero LDS scores. One potential explanation is that these functions are not well approximated by the linear functions of training samples as suggested by Park et al. (2023). Using either $\mathcal{L}_{\text{Simple}}$, $\mathcal{L}_{\text{ELBO}}$ yields reasonable LDS scores, rank different constructions of $\phi^s$ in a similar order and have similar LDS scores for $\mathcal{P}_s^\top \nabla_\theta \mathcal{L}_{\text{Simple}}$. However, setting the model output function to be $\mathcal{L}_{\text{Simple}}(\boldsymbol{x}, \theta)$ yields higher LDS scores for $\mathcal{P}_s^\top \nabla_\theta \mathcal{L}_{\text{Square}}$—30.81% and 22.62%

Table 5: LDS(%) on CIFAR-2 under #100 computation budget. The numbers in the parentheses indicate the number of timesteps and the number of noises set to compute the gradients, respectively.

| Method | Strategy | Validation | | | | Generation | | | |
|---|---|---|---|---|---|---|---|---|---|
| | | (10, 10) | (20, 5) | (50, 2) | (100, 1) | (10, 10) | (20, 5) | (50, 2) | (100, 1) |
| TRAK | cumulative | 13.45 | 14.12 | 15.71 | 17.68 | 4.58 | 5.30 | 6.52 | 8.97 |
| | uniform | 16.57 | 19.45 | 22.93 | 24.00 | 9.13 | 12.14 | 15.49 | 16.13 |
| D-TRAK | cumulative | 29.35 | 31.14 | 33.40 | 34.90 | 19.86 | 21.26 | 23.44 | 25.16 |
| | uniform | 30.71 | 32.91 | 34.06 | 34.24 | 22.07 | 24.39 | 25.76 | 26.02 |

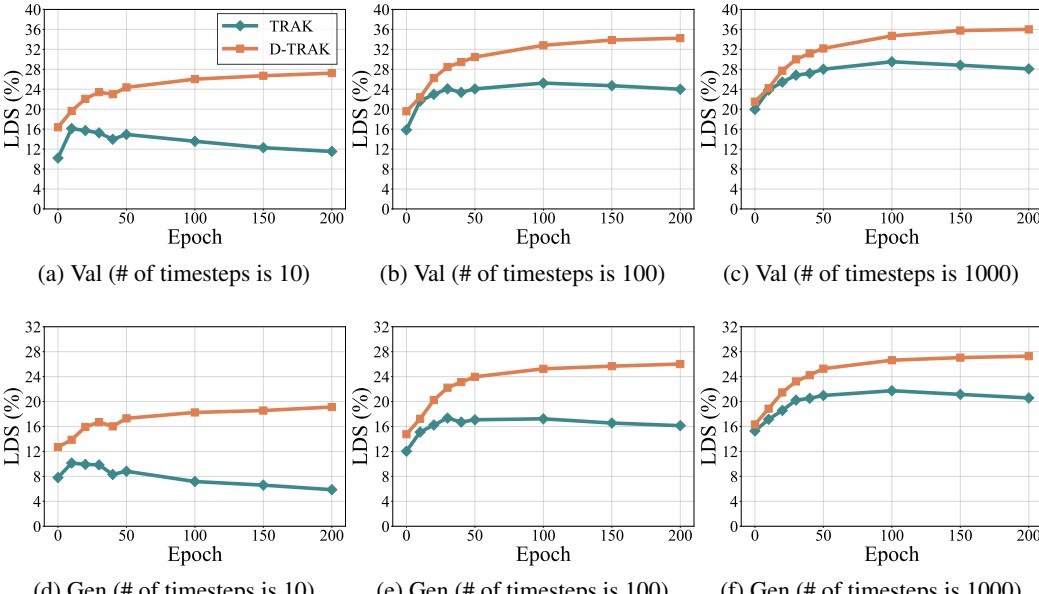

(a) Val (# of timesteps is 10)  (b) Val (# of timesteps is 100)  (c) Val (# of timesteps is 1000)

(d) Gen (# of timesteps is 10)  (e) Gen (# of timesteps is 100)  (f) Gen (# of timesteps is 1000)

Figure 6: The LDS(%) on CIFAR-2 under different checkpoints. We consider 10, 100, and 1000 timesteps selected to be evenly spaced within the interval $[1, T]$, which are used to approximate the expectation $\mathbb{E}_t$. For each sampled timestep, we sample one standard Gaussian noise $\epsilon \sim \mathcal{N}(\epsilon | \mathbf{0}, \mathbf{I})$ to approximate the expectation $\mathbb{E}_\epsilon$. We set $k = 32768$.

on validation and generation set respectively, thus we set $\mathcal{F}(\boldsymbol{x}; \theta) = \mathcal{L}_{\text{Simple}}(\boldsymbol{x}, \theta)$ as the default choice in the following experiments.

**Value of $k$.** When we compute the attribution scores, we use random projection to reduce the dimensionality of the gradients. Intuitively, as the resulting dimension $k$ increases, the associated projection better preserves inner products, but is also more expensive to compute Johnson & Lindenstrauss (1984). We thus investigate how the projection dimension $k$ selection affects the attribution performance. Figure 5 shows that the dimension is increased, and the LDS scores of both TRAK and our method initially increase and gradually saturate, as expected. In the following experiments, we set $k = 32768$ as the default choice.

**Number of noises.** When computing the gradients, for every timestep, we can sample multiple noises and then average the resulting gradients for better performance at the cost of computation. However, as shown in Table 5, increasing the number of noises is less effective than increasing the number of timesteps. So we set using one noise per timestep when computing the gradients as default.

**Timestep selection.** As we mentioned previously, we set the time selection strategy as *uniform*. We also explore another potential strategy *cumulative*. For example, when the number of timesteps is set as 10, we will select $\{1, 2, 3, \cdots, 10\}$ for *cumulative*. As shown in Table 5, for D-TRAK, the sampling strategy has little influence, while for TRAK, the *uniform* is significantly better. Thus we set the sampling strategy as *uniform* as default in the following experiments for a fair comparison.

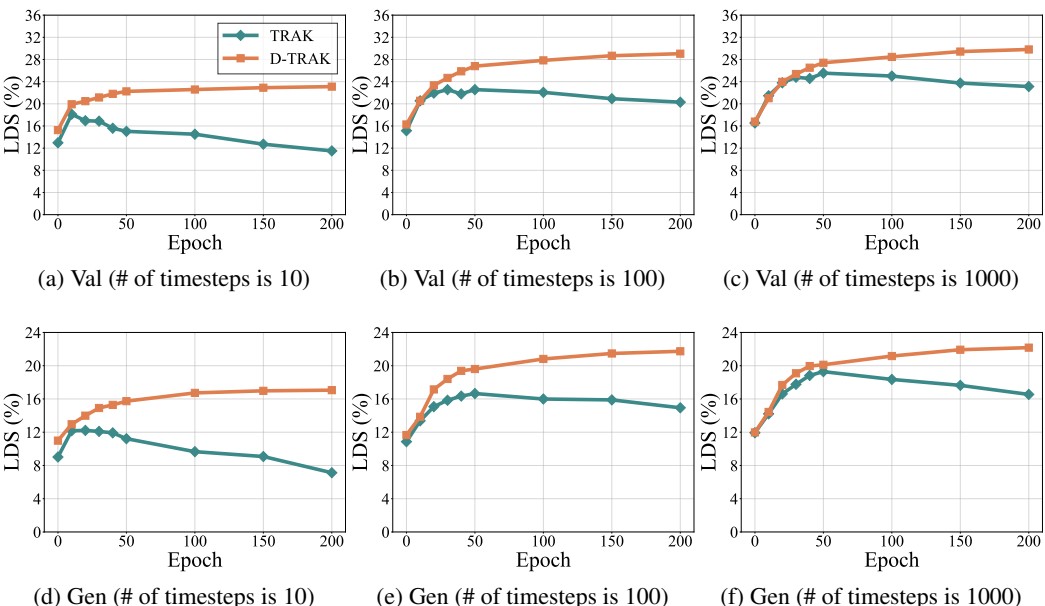

Figure 7: The LDS(%) on CelebA under different checkpoints. We consider 10, 100, and 1000 timesteps selected to be evenly spaced within the interval $[1, T]$, which are used to approximate the expectation $\mathbb{E}_t$. For each sampled timestep, we sample one standard Gaussian noise $\boldsymbol{\epsilon} \sim \mathcal{N}(\boldsymbol{\epsilon}|\mathbf{0}, \mathbf{I})$ to approximate the expectation $\mathbb{E}_{\boldsymbol{\epsilon}}$. We set $k = 32768$.

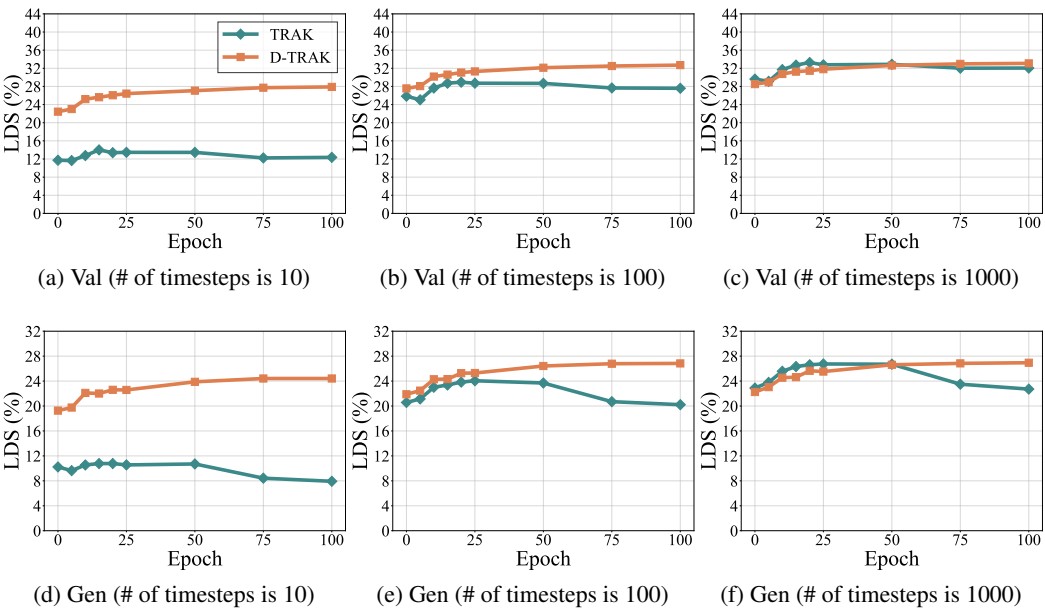

Figure 8: The LDS(%) on ArtBench-2 under different checkpoints. We consider 10, 100, and 1000 timesteps selected to be evenly spaced within the interval $[1, T]$, which are used to approximate the expectation $\mathbb{E}_t$. For each sampled timestep, we sample one standard Gaussian noise $\boldsymbol{\epsilon} \sim \mathcal{N}(\boldsymbol{\epsilon}|\mathbf{0}, \mathbf{I})$ to approximate the expectation $\mathbb{E}_{\boldsymbol{\epsilon}}$. We set $k = 32768$.

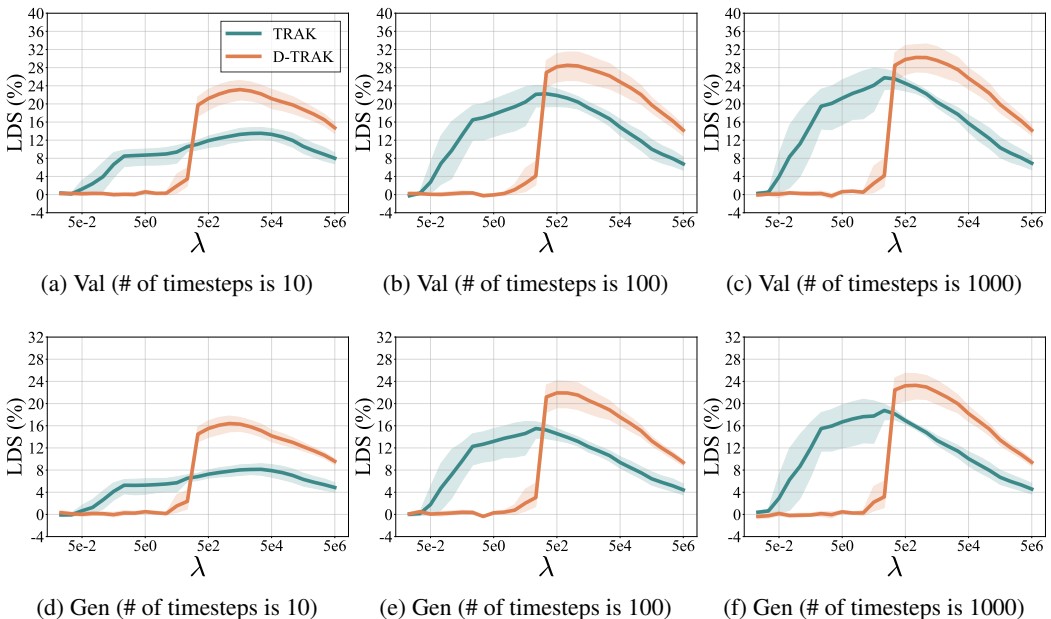

Figure 9: LDS (%) on CIFAR-2 under different $\lambda$. We consider 10, 100, and 1000 timesteps selected to be evenly spaced within the interval $[1, T]$, which are used to approximate the expectation $\mathbb{E}_t$. For each sampled timestep, we sample one standard Gaussian noise $\epsilon \sim \mathcal{N}(\epsilon|\mathbf{0}, \mathbf{I})$ to approximate the expectation $\mathbb{E}_\epsilon$. We set $k = 32768$. The shaded area in the LDS represents the standard deviation corresponding to different checkpoints.

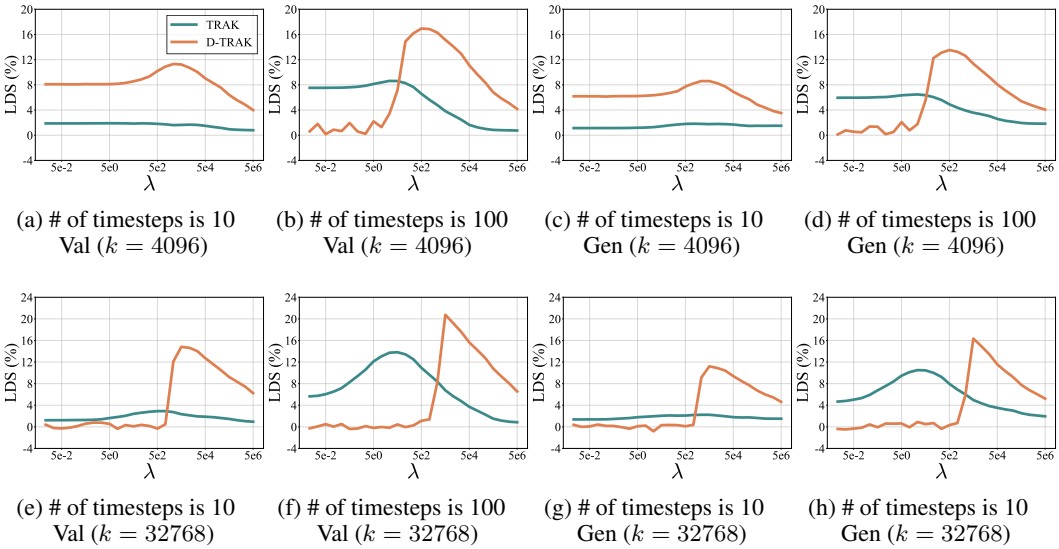

Figure 10: LDS (%) on CIFAR-10 under different $\lambda$. We consider 10 and 100 timesteps selected to be evenly spaced within the interval $[1, T]$, which are used to approximate the expectation $\mathbb{E}_t$. For each sampled timestep, we sample one standard Gaussian noise $\epsilon \sim \mathcal{N}(\epsilon|\mathbf{0}, \mathbf{I})$ to approximate the expectation $\mathbb{E}_\epsilon$. The subplots are associated with random projection dimensions 4096 and 32768.

Table 6: LDS (%) of **retraining-free methods** on CelebA with various # of timesteps (10 or 100). [†] indicates applying TRAK's scalability optimizations as described in Appendix A.3.

| Method | Validation | | Generation | |
|---|---|---|---|---|
| | 10 | 100 | 10 | 100 |
| Raw pixel (dot prod.) | 5.58 $\pm$ 0.73 | | -4.94 $\pm$ 1.58 | |
| Raw pixel (cosine) | 6.16 $\pm$ 0.75 | | -4.38 $\pm$ 1.63 | |
| CLIP similarity (dot prod.) | 8.87 $\pm$ 1.14 | | 2.51 $\pm$ 1.13 | |
| CLIP similarity (cosine) | 10.92 $\pm$ 0.87 | | 3.03 $\pm$ 1.13 | |
| Gradient (dot prod.) (Charpiat et al., 2019) | 3.82 $\pm$ 0.50 | 4.89 $\pm$ 0.65 | 3.83 $\pm$ 1.06 | 4.53 $\pm$ 0.84 |
| Gradient (cosine) (Charpiat et al., 2019) | 3.65 $\pm$ 0.52 | 4.79 $\pm$ 0.68 | 3.86 $\pm$ 0.96 | 4.40 $\pm$ 0.86 |
| TracInCP (Pruthi et al., 2020) | 5.14 $\pm$ 0.75 | 4.89 $\pm$ 0.86 | 5.18 $\pm$ 1.05 | 4.50 $\pm$ 0.93 |
| GAS (Hammoudeh & Lowd, 2022a) | 5.44 $\pm$ 0.68 | 5.19 $\pm$ 0.64 | 4.69 $\pm$ 0.97 | 3.98 $\pm$ 0.97 |
| Journey TRAK (Georgiev et al., 2023) | / | / | 6.53 $\pm$ 1.06 | 10.87 $\pm$ 0.84 |
| Relative IF[†] (Barshan et al., 2020) | 11.10 $\pm$ 0.51 | 19.89 $\pm$ 0.50 | 6.80 $\pm$ 0.77 | 14.66 $\pm$ 0.70 |
| Renorm. IF[†] (Hammoudeh & Lowd, 2022a) | 11.01 $\pm$ 0.50 | 18.67 $\pm$ 0.51 | 6.74 $\pm$ 0.82 | 13.24 $\pm$ 0.71 |
| TRAK (Park et al., 2023) | 11.28 $\pm$ 0.47 | 20.02 $\pm$ 0.49 | 7.02 $\pm$ 0.89 | 14.71 $\pm$ 0.70 |
| D-TRAK (**Ours**) | **22.83 $\pm$ 0.51** | **28.69 $\pm$ 0.44** | **16.84 $\pm$ 0.54** | **21.47 $\pm$ 0.48** |

**Checkpoint selection.** Motivated by Pruthi et al. (2020), we study the effect of using different checkpoints of a model to compute the gradients. As shown in Figures 6, 7 and 8, for D-TRAK, the best LDS scores are obtained at the final checkpoint. However, for TRAK, using the final checkpoint is not the best choice. Finding which checkpoint yields the best LDS score requires computing the attributions many times, which means much more computational cost. Additionally, in practice, we might only get access to the final model.

**Number of timesteps.** When computing the attribution scores, considering the gradients from more timesteps might increase the performance. That is, there is a trade-off between efficacy and computational efficiency. We consider the extreme case that we compute gradients for all the 1000 timesteps, which is computationally expensive. As shown in Figures 6, 7 and 8, the more timestep we consider, the better LDS we can get. However, for our method, setting the number of timesteps as 100 even 10 could get performance similar to 1000, while being much faster.

**Value of $\lambda$.** In our initial experiments, we noticed that adding $\lambda I$ to the kernel before inverting the kernel matrix, is necessary for obtaining better LDS scores. This term may serve for numerical stability and regularization effect as suggested by Hastie (2020). Traditionally the $\lambda$ is small and close to zero, however, in our case, we found that we need to set a relatively larger $\lambda$, which might be because the kernel is not divided by the number of training samples $N$ following Park et al. (2023). We search the $\lambda$ from {1e-2, 2e-2, 5e-2, 1e-1, 2e-1, 5e-1, $\cdots$, 1e6, 2e6, 5e6}. As shown in Figures 9 and 10, setting a proper $\lambda$, which adjusts the regularization level, has a significant influence on the attribution performance. In our experiments, we search $\lambda$ on the LDS benchmarks directly and report the peak LDS scores for both TRAK and D-TRAK for a fair comparison. In practice, we can search the proper $\lambda$ value by manually examining the attribution visualizations or evaluating the LDS scores on a smaller scale LDS benchmark.

## C ADDITIONAL EXPERIMENT RESULTS

In this section, we provide additional experiment results of LDS evaluation (on both retraining-free methods and retraining-based methods) and counterfactual evaluation.

### C.1 LDS EVALUATION (RETRAINING-FREE METHODS)

As shown in Table 6, on the validation set, with 10/100 timesteps, D-TRAK achieves improvements of +11.55%/+8.67% on CelebA. On the generation set, with 10/100 timesteps, D-TRAK exhibits gains of +9.82%/+6.76% in the LDS scores. These results highlight the efficacy of D-TRAK and underscore its capacity for enhancing LDS scores.

Table 7: LDS (%) of **retraining-based methods** on CIFAR-2 and ArtBench-2 with various $S$. For TRAK and D-TRAK, we select 10 and 100 timesteps evenly spaced within the interval $[1, T]$ to approximate $\mathbb{E}_t$. For each selected timestep, we sample one standard Gaussian noise to approximate $\mathbb{E}_\epsilon$. We set $k = 32768$.

| | | Validation | | Generation | |
|---|---|---|---|---|---|
| **Results on CIFAR-2** | | | | | |
| Method | $S$ | 10 | 100 | 10 | 100 |
| TRAK (ensemble) (Park et al., 2023) | 1 | $12.85 \pm 0.41$ | $25.52 \pm 0.43$ | $6.91 \pm 0.44$ | $17.50 \pm 0.43$ |
| | 2 | $13.19 \pm 0.43$ | $27.29 \pm 0.43$ | $7.00 \pm 0.41$ | $19.23 \pm 0.42$ |
| | 4 | $13.27 \pm 0.43$ | $27.98 \pm 0.43$ | $7.08 \pm 0.41$ | $20.12 \pm 0.41$ |
| | 8 | $13.36 \pm 0.42$ | $28.59 \pm 0.43$ | $7.13 \pm 0.41$ | $20.67 \pm 0.41$ |
| D-TRAK (**Ours**) (ensemble) | 1 | $25.71 \pm 0.37$ | $32.26 \pm 0.28$ | $18.90 \pm 0.47$ | $24.92 \pm 0.42$ |
| | 2 | $27.12 \pm 0.38$ | $34.09 \pm 0.32$ | $20.05 \pm 0.49$ | $26.61 \pm 0.43$ |
| | 4 | $27.77 \pm 0.36$ | $34.88 \pm 0.31$ | $20.59 \pm 0.46$ | $27.30 \pm 0.45$ |
| | 8 | $\mathbf{28.07 \pm 0.32}$ | $\mathbf{35.42 \pm 0.33}$ | $\mathbf{20.82 \pm 0.44}$ | $\mathbf{27.75 \pm 0.43}$ |
| Empirical IF (Feldman & Zhang, 2020) | 64 | $6.00 \pm 0.50$ | | $4.12 \pm 0.46$ | |
| | 128 | $8.83 \pm 0.43$ | | $6.10 \pm 0.43$ | |
| | 256 | $12.17 \pm 0.47$ | | $9.47 \pm 0.55$ | |
| | 512 | $16.69 \pm 0.51$ | | $12.97 \pm 0.47$ | |
| Datamodel (Ilyas et al., 2022) | 64 | $5.99 \pm 0.50$ | | $4.08 \pm 0.45$ | |
| | 128 | $8.86 \pm 0.44$ | | $6.08 \pm 0.45$ | |
| | 256 | $12.25 \pm 0.46$ | | $9.55 \pm 0.54$ | |
| | 512 | $16.94 \pm 0.50$ | | $12.81 \pm 0.41$ | |
| **Results on ArtBench-2** | | | | | |
| Method | $S$ | 10 | 100 | 10 | 100 |
| TRAK (ensemble) (Park et al., 2023) | 1 | $12.84 \pm 0.46$ | $28.20 \pm 0.33$ | $9.31 \pm 0.51$ | $21.32 \pm 0.77$ |
| | 2 | $14.86 \pm 0.45$ | $31.24 \pm 0.36$ | $11.19 \pm 0.51$ | $24.12 \pm 0.70$ |
| | 4 | $16.09 \pm 0.49$ | $32.89 \pm 0.34$ | $12.16 \pm 0.56$ | $25.63 \pm 0.65$ |
| | 8 | $16.63 \pm 0.49$ | $33.91 \pm 0.36$ | $12.88 \pm 0.62$ | $26.50 \pm 0.65$ |
| D-TRAK (**Ours**) (ensemble) | 1 | $27.39 \pm 0.40$ | $31.99 \pm 0.35$ | $23.51 \pm 0.58$ | $25.68 \pm 0.70$ |
| | 2 | $28.78 \pm 0.42$ | $33.08 \pm 0.36$ | $24.18 \pm 0.63$ | $26.42 \pm 0.72$ |
| | 4 | $29.70 \pm 0.44$ | $33.84 \pm 0.39$ | $24.87 \pm 0.67$ | $26.94 \pm 0.75$ |
| | 8 | $\mathbf{30.21 \pm 0.44}$ | $\mathbf{34.24 \pm 0.39}$ | $\mathbf{25.09 \pm 0.69}$ | $\mathbf{27.20 \pm 0.76}$ |
| Empirical IF (Feldman & Zhang, 2020) | 64 | $4.91 \pm 0.50$ | | $5.29 \pm 0.75$ | |
| | 128 | $7.62 \pm 0.49$ | | $8.67 \pm 0.91$ | |
| | 256 | $11.27 \pm 0.44$ | | $11.51 \pm 0.68$ | |
| | 512 | $16.10 \pm 0.43$ | | $16.45 \pm 0.67$ | |
| Datamodel (Ilyas et al., 2022) | 64 | $4.99 \pm 0.51$ | | $5.30 \pm 0.76$ | |
| | 128 | $7.83 \pm 0.46$ | | $8.70 \pm 0.88$ | |
| | 256 | $11.32 \pm 0.44$ | | $11.48 \pm 0.62$ | |
| | 512 | $16.19 \pm 0.47$ | | $16.35 \pm 0.69$ | |

## C.2 LDS EVALUATION (RETRAINING-BASED METHODS)

As shown in Tables 7 and 8, we compare D-TRAK with those retraining-based methods as references on CIFAR-2, CelebA, and ArtBench-2. Under this setting, we need to retrain multiple models on different random subsets of the full training set and compute gradients multiple times, where each subset has a fixed size of $\beta \cdot N$. We set $\beta = 0.5$. At the cost of computation, using the gradients computed by independently trained models improves the LDS scores for both TRAK and D-TRAK. Considering 8 models, D-TRAK still performs better than TRAK on various settings.

We also compare D-TRAK with both Empirical Influence and Datamodel. These two methods are computationally expensive so we train at most 512 models due to our computation limits. Overall, these two methods perform similarly on our settings. With larger $S$, which translates into retraining

Table 8: LDS (%) of **retraining-based methods** on CelebA with various $S$. For TRAK and D-TRAK, we select 10 and 100 timesteps evenly spaced within the interval $[1, T]$ to approximate $\mathbb{E}_t$. For each selected timestep, we sample one standard Gaussian noise to approximate $\mathbb{E}_\epsilon$. We set $k = 32768$.

| Method | $S$ | Validation | | Generation | |
|---|---|---|---|---|---|
| | | 10 | 100 | 10 | 100 |
| TRAK (ensemble) (Park et al., 2023) | 1 | $13.07 \pm 0.44$ | $22.14 \pm 0.45$ | $9.37 \pm 0.75$ | $16.51 \pm 0.51$ |
| | 2 | $13.50 \pm 0.43$ | $23.73 \pm 0.44$ | $9.26 \pm 0.73$ | $17.38 \pm 0.58$ |
| | 4 | $13.52 \pm 0.44$ | $24.62 \pm 0.42$ | $9.38 \pm 0.69$ | $18.01 \pm 0.57$ |
| | 8 | $13.64 \pm 0.42$ | $24.93 \pm 0.40$ | $9.36 \pm 0.68$ | $18.37 \pm 0.58$ |
| D-TRAK (**Ours**) (ensemble) | 1 | $23.04 \pm 0.56$ | $27.90 \pm 0.52$ | $17.15 \pm 0.59$ | $20.87 \pm 0.48$ |
| | 2 | $23.80 \pm 0.56$ | $28.72 \pm 0.53$ | $17.60 \pm 0.59$ | $21.79 \pm 0.52$ |
| | 4 | $24.08 \pm 0.55$ | $29.57 \pm 0.49$ | $17.74 \pm 0.59$ | $22.38 \pm 0.48$ |
| | 8 | $\mathbf{24.19 \pm 0.53}$ | $\mathbf{29.64 \pm 0.50}$ | $\mathbf{17.87 \pm 0.57}$ | $\mathbf{22.58 \pm 0.49}$ |
| Empirical IF (Feldman & Zhang, 2020) | 64 | $6.44 \pm 0.54$ | | $4.58 \pm 0.42$ | |
| | 128 | $8.23 \pm 0.54$ | | $5.85 \pm 0.44$ | |
| | 256 | $11.07 \pm 0.43$ | | $8.82 \pm 0.36$ | |
| | 512 | $15.55 \pm 0.42$ | | $12.62 \pm 0.38$ | |
| Datamodel (Ilyas et al., 2022) | 64 | $6.41 \pm 0.55$ | | $4.51 \pm 0.42$ | |
| | 128 | $8.16 \pm 0.55$ | | $5.94 \pm 0.44$ | |
| | 256 | $11.14 \pm 0.44$ | | $8.92 \pm 0.38$ | |
| | 512 | $15.56 \pm 0.43$ | | $12.43 \pm 0.36$ | |

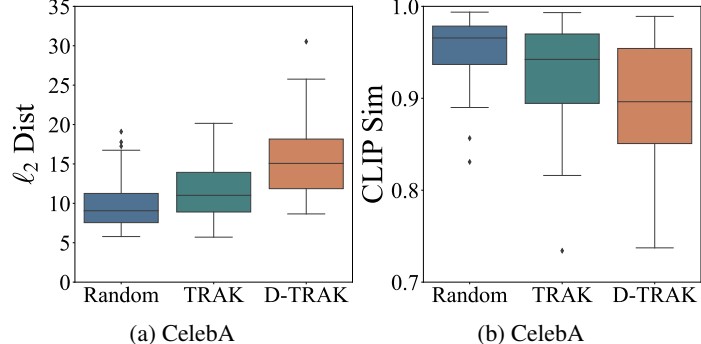

(a) CelebA        (b) CelebA

Figure 11: Boxplots of counterfactual evaluation on CelebA. We quantify the impact of removing the 1,000 highest-scoring training samples and re-training according to Random, TRAK, and D-TRAK. We measure the pixel-wise $\ell_2$-distance and CLIP cosine similarity between 60 synthesized samples and corresponding images generated by the re-trained models, sampled from the same random seed.

more models and thus more computation needed, the LDS scores grow gradually. However, using 512 models, the LDS scores obtained by these two methods are still substantially lower than D-TRAK, even TRAK.

## C.3 COUNTERFACTUAL EVALUATION

In the context of CelebA, the counterfactual evaluation of D-TRAK against TRAK, as presented in Figure 11, demonstrates that D-TRAK can better identify influential images that have a larger impact on the generated images. When examining the median pixel-wise $\ell_2$ distance resulting from removing-and-retraining, D-TRAK yields 15.07, in contrast to TRAK's values of 11.02. A reverse trend is observed when evaluating the median CLIP cosine similarity, where D-TRAK obtains lower similarities of 0.896, which are notably lower than TRAK's 0.942.

Random TRAK D-TRAK   Random TRAK D-TRAK

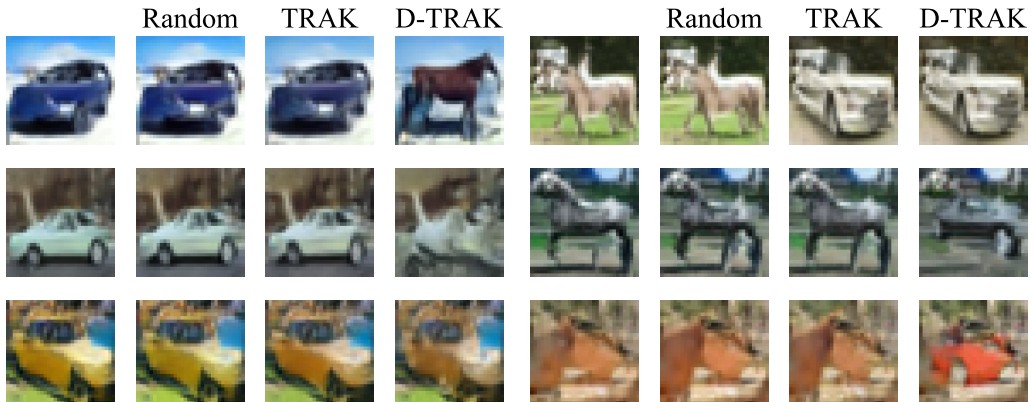

Figure 12: Counterfactual visualization of (**Left**) automobile and (**Right**) horse on CIFAR-2. We compare the original generated samples to those generated from the same random seed with the retrained models.

Random TRAK D-TRAK   Random TRAK D-TRAK

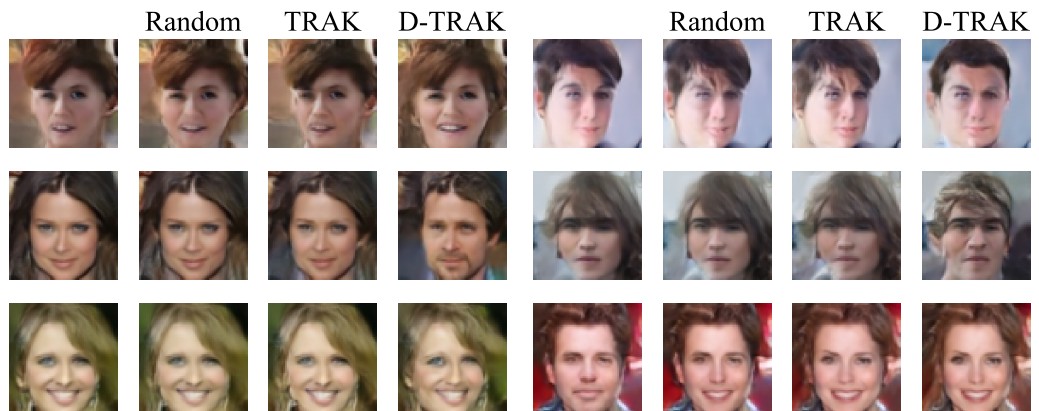

Figure 13: Counterfactual visualization (**Left**) female and (**Right**) male on CelebA. We compare the original generated samples to those generated from the same random seed with the retrained models.

## D Visualization

We provide more visualization results, including counterfactual visualization, proponents and opponents visualization, as well as self-influence (memorization) visualization.

### D.1 Counterfactual visualization

In the context of CIFAR-2, CelebA, and ArtBench-2 datasets, we compare the original generated samples to those generated from the same random seed with the models trained after the exclusion of the top 1,000 positive influencers identified by different attribution methods including Random, TRAK, and D-TRAK. For both D-TRAK and TRAK, we consider 100 timesteps selected to be evenly spaced within the interval $[1, T]$, sample one standard Gaussian noise, and set $k = 32768$ when computing the gradients. In terms of the LDS(%) performance on the generation sets of CIFAR-2, CelebA, and ArtBench-2, D-TRAK and TRAK obtain 25.67% versus 15.87%, 21.47% versus 14.71%, and 26.53% versus 20.02% respectively, as reported previously.

As shown in Figure 12, our results suggest that our method D-TRAK can better identify influential images that have a significant impact on the target image. Take the first row of the left column in

Figure 12 as an example, the model retrained after removing training samples based on Random and TRAK still generates a *automobile* image, while the one corresponding to D-TRAK generates an image looks similar to a mixture of *automobile* and *horse*. It is also interesting to see that for CIFAR-2 and CelebA, randomly removing 1,000 training samples results in little change in the generated images based on manual perception.

## D.2 PROPONENTS AND OPPONENTS VISUALIZATION

Following Pruthi et al. (2020), we term training samples that have a positive influence score as proponents and samples that have a negative value of influence score as opponents. For samples of interest, we visualize the training samples corresponding to the most positive and negative attribution scores. The visualizations on CIFAR-2, CIFAR-10, CelebA, ArtBench-2 and ArtBench-5 are shown in Figures 15, 16, 17, 18 and 19, respectively. We manually check the proponents and opponents. We observe that D-TRAK consistently finds proponents visually more similar to the target samples.

## D.3 SELF-INFLUENCE (MEMORIZATION) VISUALIZATION

Motivated by Feldman & Zhang (2020); Zheng & Jiang (2022); Gu et al. (2023), we also visualize the training samples that have the highest self-influence scores on CIFAR-2 and ArtBench-2 as shown in Figure 20. The self-influence of $x^n$ is computed as $\tau_{\text{D-TRAK}}(x^n, \mathcal{D})_n$. When computing the self-influence scores, we consider 100 timesteps selected to be evenly spaced within the interval $[1, T]$, and set $k = 32768$. The identified highly self-influenced samples usually look atypical or unique, while low self-influence ones have similar samples in the training set. Especially, on CIFAR-2, highly self-influenced samples are visually high-contrast.

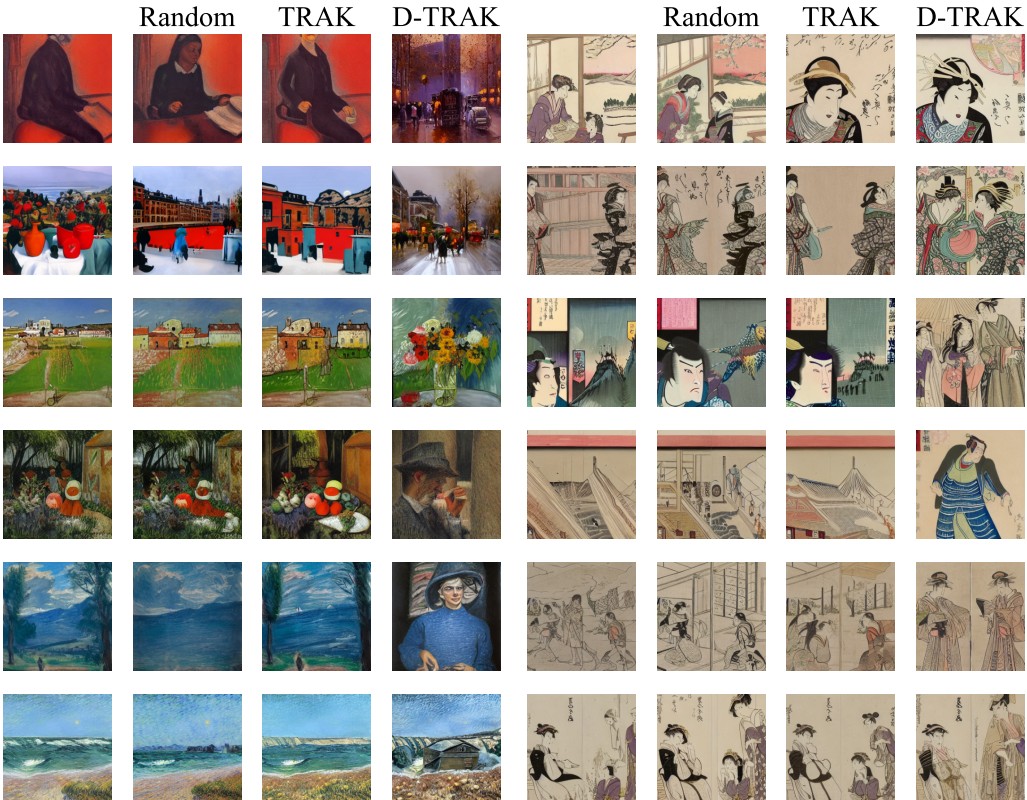

Figure 14: Counterfactual visualization (**Left**) post-imporessaion paintings and (**Right**) ukiyo-e paintings on ArtBench-2. We compare the original generated samples to those generated from the same random seed with the retrained models.

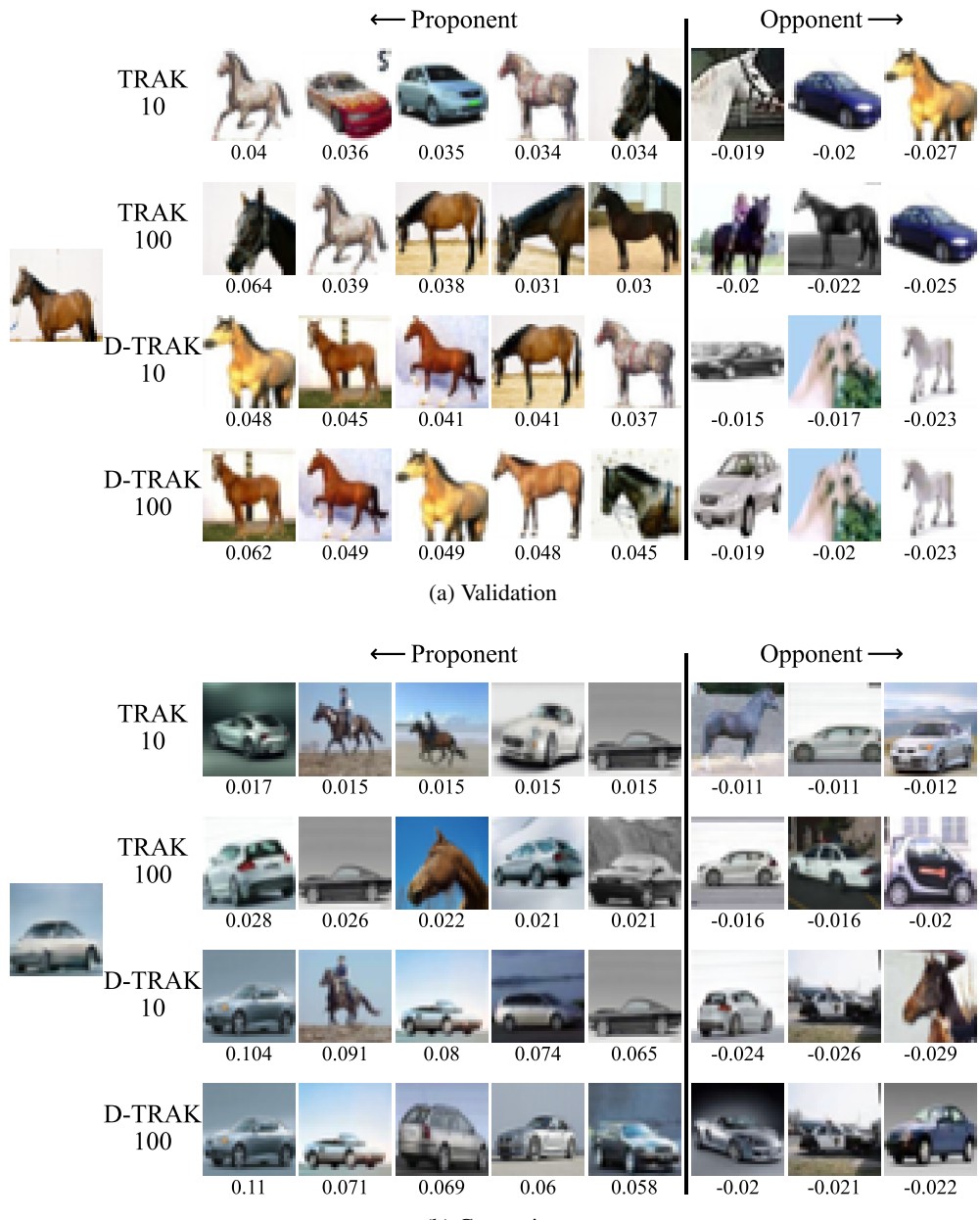

Figure 15: Proponents and opponents visualization on CIFAR-2 using TRAK and D-TRAK with various # of timesteps (10 or 100). For each sample of interest, 5 most positive influential training samples and 3 most negative influential training samples are given together with the influence scores (below each sample).

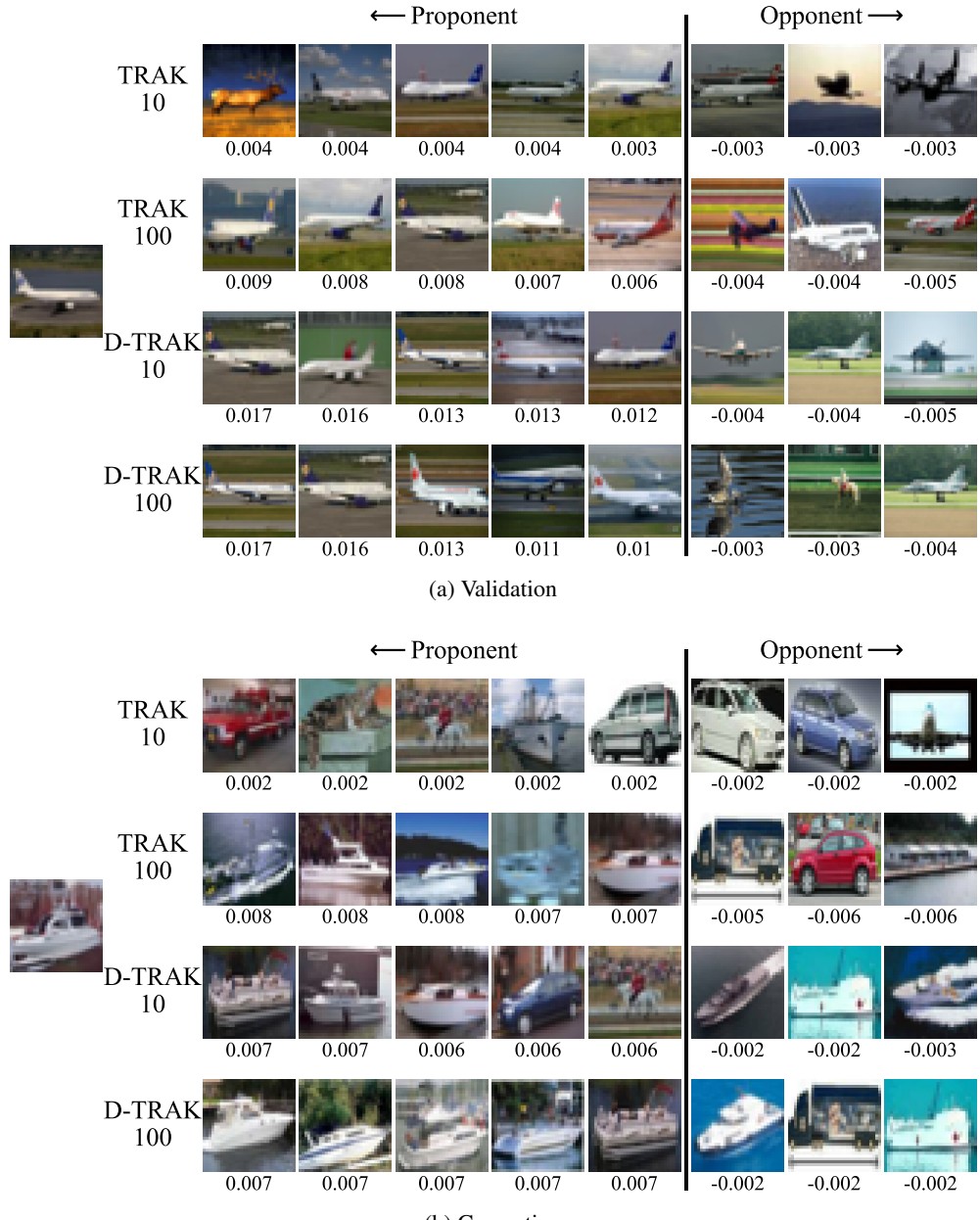

Figure 16: Proponents and opponents visualization on CIFAR-10 using TRAK and D-TRAK with various # of timesteps (10 or 100). For each sample of interest, 5 most positive influential training samples and 3 most negative influential training samples are given together with the influence scores (below each sample).

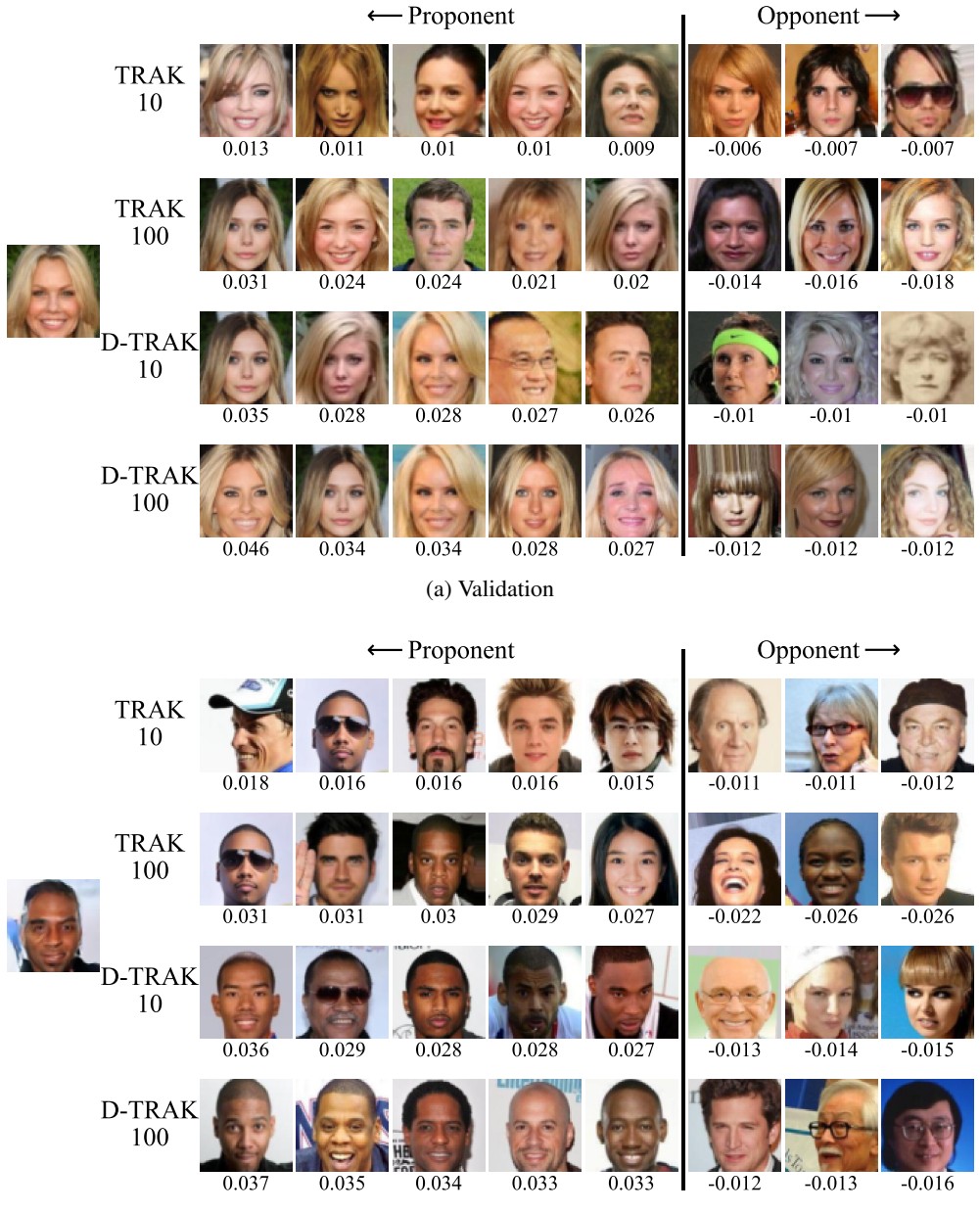

Figure 17: Proponents and opponents visualization on CelebA using TRAK and D-TRAK with various # of timesteps (10 or 100). For each sample of interest, 5 most positive influential training samples and 3 most negative influential training samples are given together with the influence scores (below each sample).

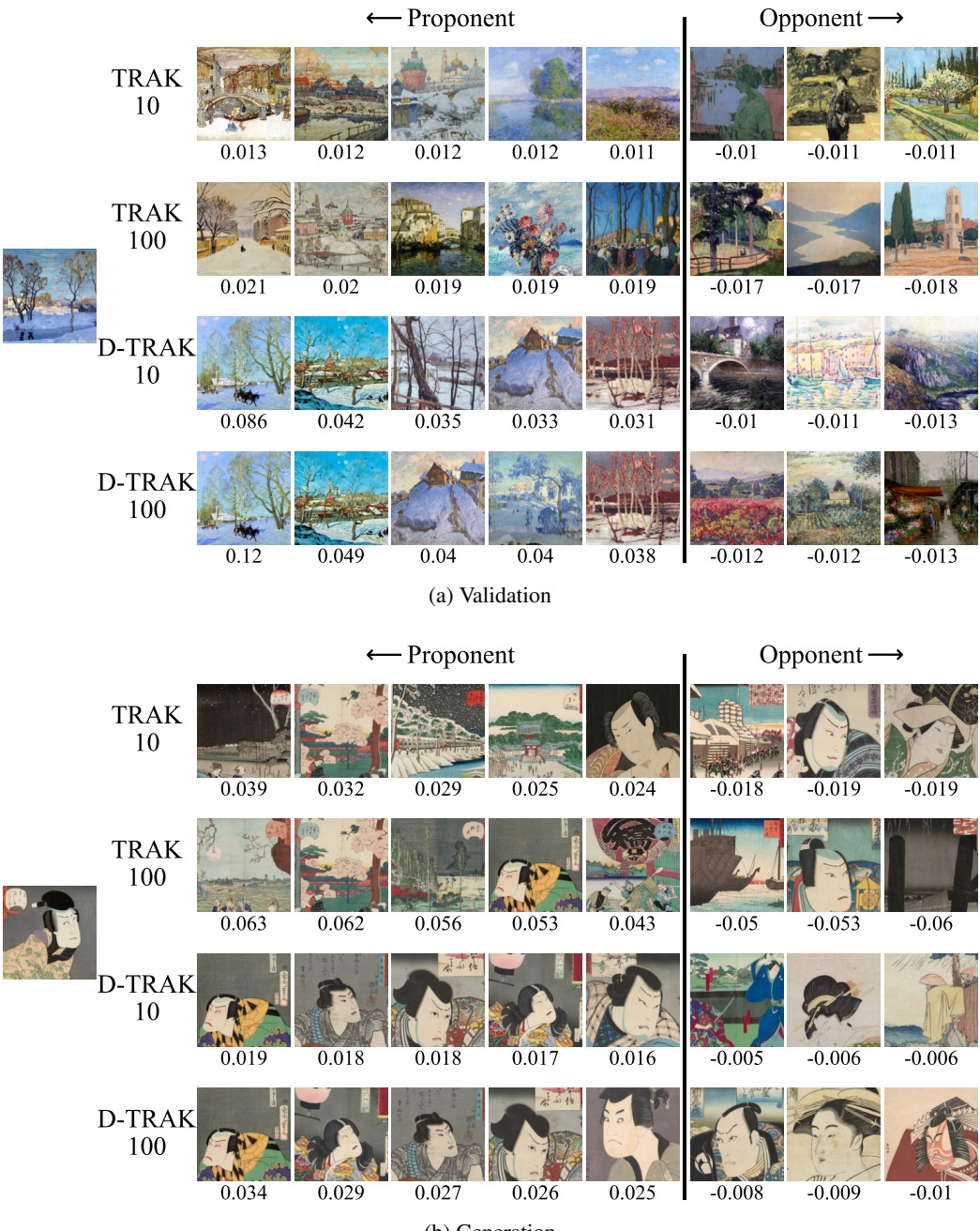

Figure 18: Proponents and opponents visualization on ArtBench-2 using TRAK and D-TRAK with various # of timesteps (10 or 100). For each sample of interest, 5 most positive influential training samples and 3 most negative influential training samples are given together with the influence scores (below each sample).

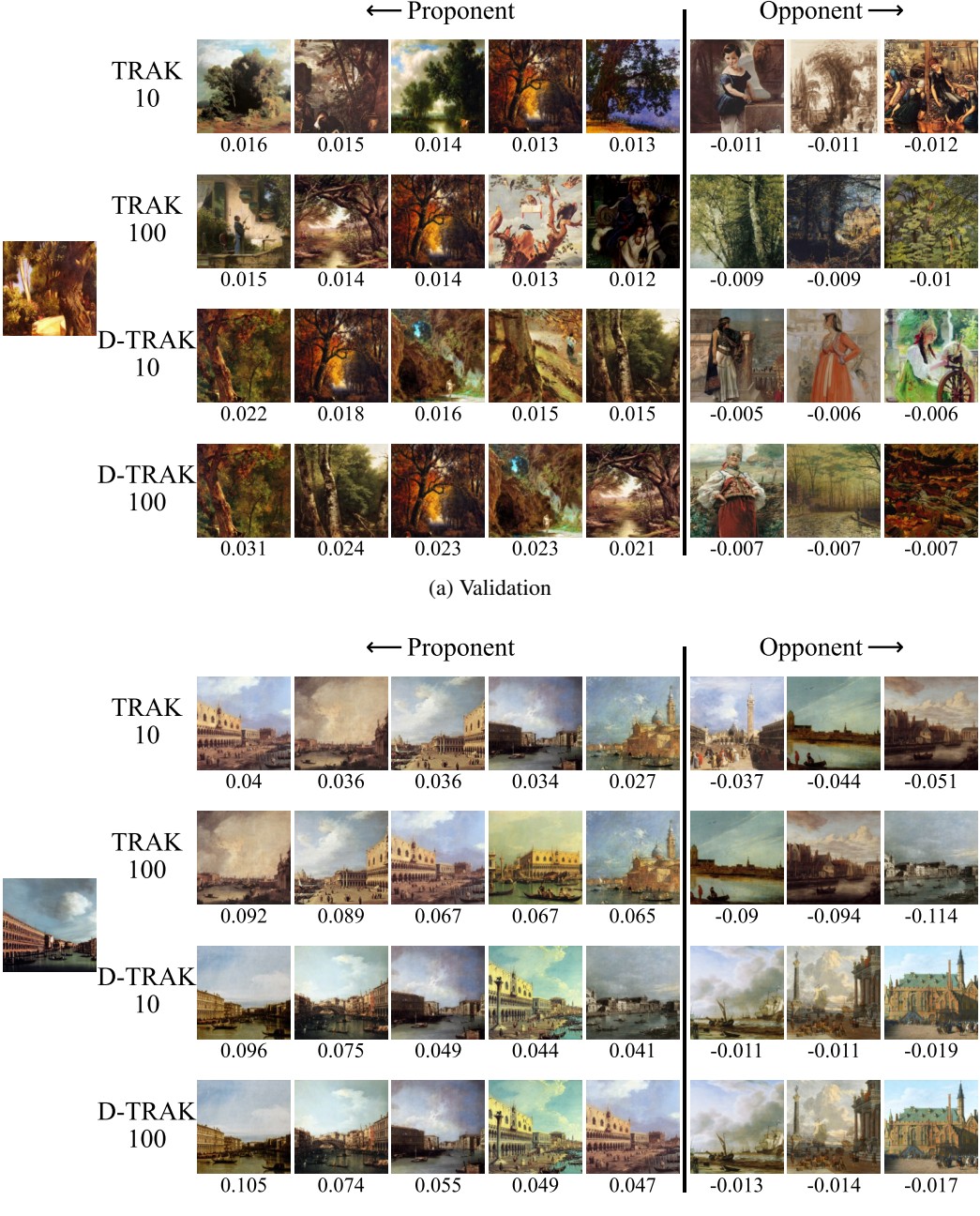

Figure 19: Proponents and opponents visualization on ArtBench-5 using TRAK and D-TRAK with various # of timesteps (10 or 100). For each sample of interest, 5 most positive influential training samples and 3 most negative influential training samples are given together with the influence scores (below each sample).

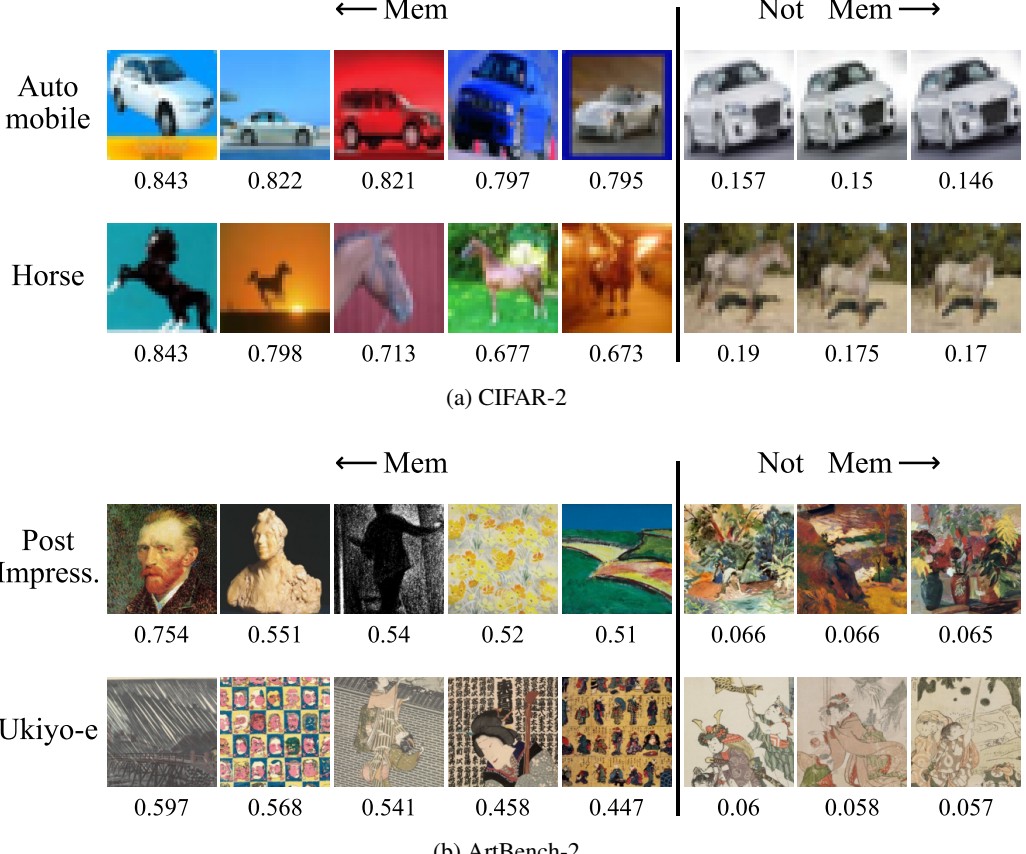

Figure 20: Self-influence visualization on CIFAR-2 and ArtBench-2. For each class, the top 5 self-influenced training samples and the bottom 3 self-influenced ones are given together with the self-influence scores (below each sample). Samples with high self-influence scores look more unique, while those with low self-influence scores have similar samples in the training set.

# E   COUNTERFACTUAL EVALUATION AT A PARTICULAR STEP

Journey TRAK Georgiev et al. (2023) is designed to conduct data attribution at a particular sampling timestep and thus produce different attribution scores for each timestep. However, our method D-TRAK is designed to conduct data attribution for the final generated images and thus produce only one attribution for the entire diffusion trajectory. To conduct a counterfactual evaluation between these two methods, we adapt D-TRAK to Journey TRAK's setting by **sharing the same attribution scores for all timesteps**. We also apply this adaptation to the Random and TRAK baselines and include them in this evaluation.

We consider timestep 400 and 300. Then we compute attribution scores on 60 generated samples using different attribution methods. For Random, TRAK, and D-TRAK, we still use the retraining-free settings described in Section 4.4 and share the same attribution scores for both timestep 400 and 300. For Journey TRAK, we compute the attribution scores specific to each timestep. Especially, for Journey TRAK, we set the number of resampling rounds of the random noise at each timestep as 10 when computing the gradients with respect to Journey TRAK's model output function. We also consider an ensemble variant of Journey TRAK to strengthen its performance via ensembling the attribution scores computed from 8 models. Given the attribution scores for each sample, we then retrain the model after removing the corresponding top-1000 influencers.

For simplicity, we abbreviate Journey TRAK as J-TRAK and the ensemble variant as J-TRAK (8). As shown in Figures 21 and 23, when examining the median pixel-wise $\ell_2$-dist distance resulting from removing-and-retraining, D-TRAK yields 7.39/8.71 and 194.18/195.72 for CIFAR-2 and ArtBench-2 at timestep 400/300 respectively, in contrast to J-TRAK (8)'s values of 6.63/7.01 and 174.76/175.06. D-TRAK obtains median similarities of 0.927/0.899 and 0.773/0.740 for the above two datasets at timestep 400/300 respectively, which are notably lower than J-TRAK (8)'s values of 0.961/0.939 and 0.816/0.805, highlighting the effectiveness of our method.

We manually compare the original generated samples to those generated from the same random seed with the re-trained models. As shown in Figures 22 and 24, our results first show that J-TRAK (8) can successfully induce a larger effect on diffusion models' generation results via removing high-scoring training images specific to a sampling timestep, compared to baselines like Random and TRAK. Second, although D-TRAK is not designed for attributing a particular sampling timestep, our results suggest our method can better identify images that have a larger impact on the target image at a specific timestep via removing high-scoring training images based on attribution scores derived from the entire sampling trajectory. Finally, it is also worth noting that for different timesteps, J-TRAK (8) identifies different influential training examples to remove and thus leads to different generation results. Take the 4th row in both Figures 22 and 24 as an example, for the timestep 400, J-TRAK (8) leads the diffusion model to generate a man with the white beard. However, for the timestep 300, it is a man with brown bread and an extra hat. In contrast, D-TRAK removes the same set of influential training examples for different timesteps and thus produces a consistent generation: a still-life painting for both timestep 400 and 300. The above phenomenon again highlights the differences between Journey TRAK and D-TRAK.

Journey TRAK may perform better via ensembling more models like 50 models at the cost of computations as suggested in Georgiev et al. (2023). Nonetheless, in this paper, we are more interested in the retraining-free setting.

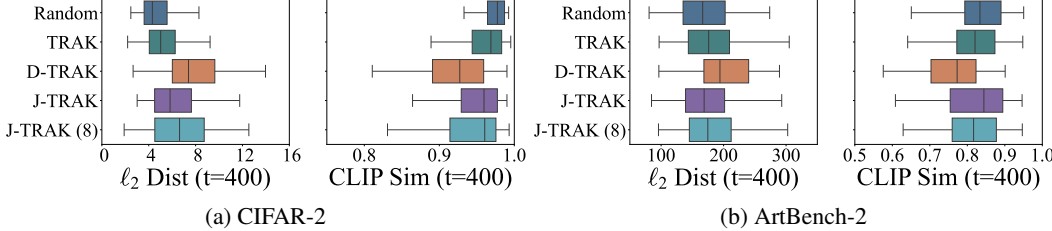

(a) CIFAR-2          (b) ArtBench-2

Figure 21: Boxplots of counterfactual evaluation **at timestep 400** on CIFAR-2 and ArtBench-2. We quantify the impact of removing the 1,000 highest-scoring training samples and re-training according to Random, TRAK, D-TRAK, J-TRAK and J-TRAK (8). J-TRAK is short for Journey TRAK. We measure the pixel-wise $\ell_2$-distance and CLIP cosine similarity between 60 synthesized samples and corresponding images generated by the re-trained models when sampling from the same random seed.

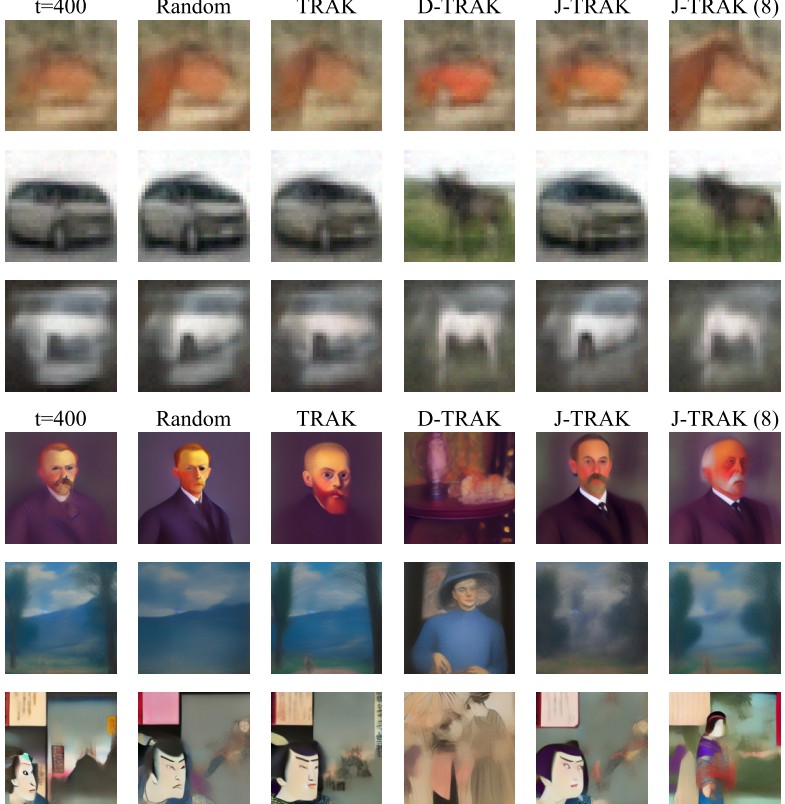

Figure 22: Counterfactual visualization **at timestep 400** on CIFAR-2 and ArtBench-2. We compare the original generated samples to those generated from the same random seed with the retrained models. The images are blurry as expected since they are noisy generated images at a particular timestep.

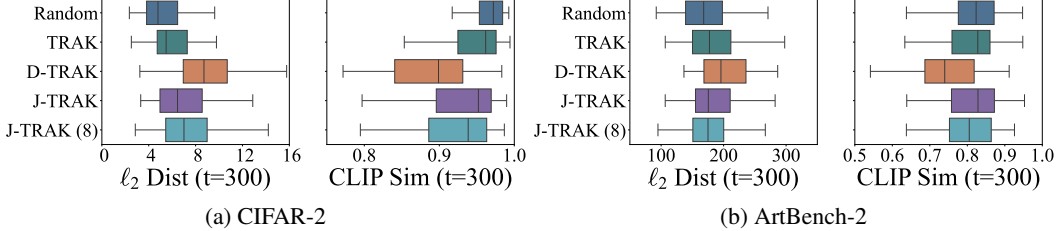

(a) CIFAR-2                                        (b) ArtBench-2

Figure 23: Boxplots of counterfactual evaluation **at timestep 300** on CIFAR-2 and ArtBench-2. We quantify the impact of removing the 1,000 highest-scoring training samples and re-training according to Random, TRAK, D-TRAK, J-TRAK and J-TRAK (8). J-TRAK is short for Journey TRAK. We measure the pixel-wise $\ell_2$-distance and CLIP cosine similarity between 60 synthesized samples and corresponding images generated by the re-trained models when sampling from the same random seed.

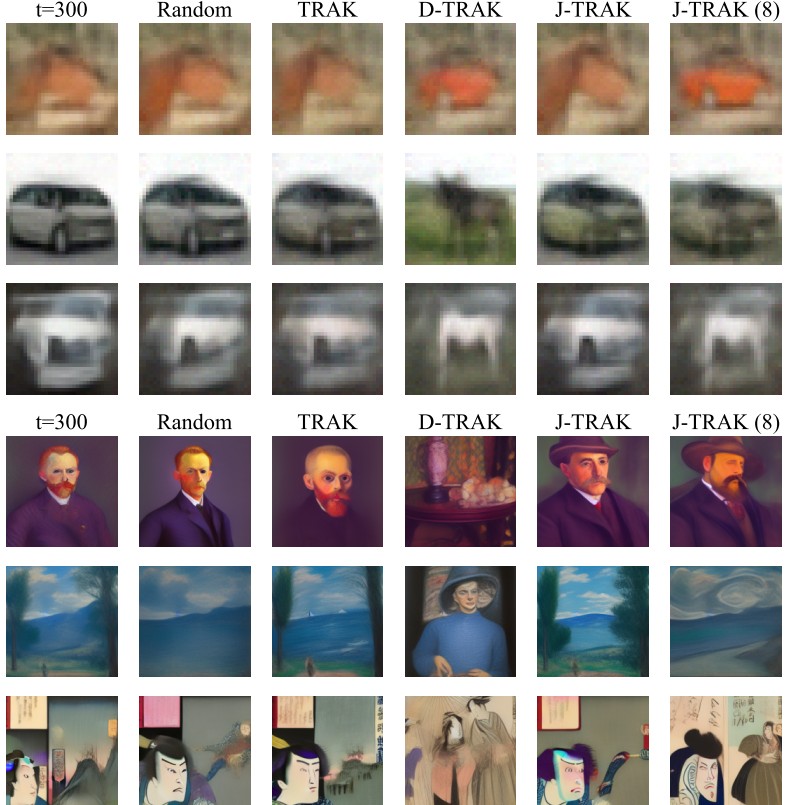

Figure 24: Counterfactual visualization **at timestep 300** on CIFAR-2 and ArtBench-2. We compare the original generated samples to those generated from the same random seed with the retrained models. The images are blurry as expected since they are noisy generated images at a particular timestep.

