# OpenReview forum: "Intriguing Properties of Data Attribution on Diffusion Models"
_ICLR.cc/2024/Conference — ICLR 2024 poster_

### Official Review · Reviewer_3yAm · 2023-11-01

**Soundness:** 2 fair
**Presentation:** 3 good
**Contribution:** 3 good
**Rating:** 6
**Confidence:** 3

**Summary:**

This work considers the topic of data attribution with diffusion models, which means quantifying the importance of training examples to model generations. Previous work on this topic has explored a range of techniques, including with gradient-based calculations and retraining-based calculations, and this work builds off of TRAK (a gradient-based method that can leverage retraining runs, but can also avoid doing so).

The proposed method, D-TRAK, deviates from TRAK in ways that the authors describe as theoretically unjustified. Namely, they use a non-standard measure of parameter sensitivity for each generation, and they omit a term that should only be ignored when the loss and model output $\mathcal{F}$ are identical. Surprisingly, they find that this heuristic version of TRAK works better than the original version, achieving better performance in the LDS metric that's designed around TRAK's original formulation.

This raises natural questions about why, which the authors don't answer. However, they observe consistently better performance than competing methods across several datasets, a couple evaluation approaches, and that performance improves when several retraining runs are used.

**Strengths:**

Data attribution is an interesting problem that's increasingly important with widely used generative models trained on web-scraped datasets. It's computationally and theoretically challenging, so developing new methods for this task is a valuable contribution. This work builds off of one the better-performing methods in the literature, TRAK, and observes performance for D-TRAK that makes it, to my knowledge, the most effective method available. And it preserves the advantages of TRAK, particularly the relative efficiency compared to methods that require many model retraining runs (e.g., the original datamodels approach from which TRAK is derived).

The evaluation is thorough, although it's restricted mostly to small datasets. This is perhaps understandable given the cost of running certain retraining based methods, but it would be helpful for the authors to clarify why they can't use ArtBench itself, for example. It seems like the cost of running TRAK/D-TRAK should be manageable, given that retraining is optional in these methods?

The experiments are also thorough in terms of the baselines considered. I appreciated the appendix section that concisely describes each method.

**Weaknesses:**

The main weakness is that in terms of data attribution methodology, the contribution here is shallow. The paper essentially finds that a couple heuristics improve performance, and offers no explanation for why. The paper acknowledges this with statements like "the mechanism of data attribution requires a deeper understanding," which are true, but this is not ideal for a publication. A paper proposing a new and improved method should offer some understanding of why it works, and the paper barely attempts to do so. The analysis in Section 3.2 about interpolating between TRAK and D-TRAK provides no insight on why D-TRAK works, it only shows that interpolated versions are strictly worse (which is not an especially interesting point, I don't see why we would have expected this to work).

The point above is my main concern, and I wonder whether this paper could be more valuable given time to revise it and offer an appropriate explanation for why D-TRAK works.

Several other thoughts:

- There is an emphasis in the introduction on the role of non-convexity in this setting, and its potential impact on TRAK not working as expected. After reading the paper, I'm not sure it ultimately addresses this point, or offers any explanation for how D-TRAK circumvents the issue. Can the authors expand on that subject in the paper?

- In Section 2.2, I'm not sure it's correct that Shapley values are proposed to *evaluate* data attributions, they're suggested to define data attributions (or more specifically, data valuation scores). In any case, this would be due to Ghorbani & Zou, not Lundberg & Lee.

- In Section 2.2, the lead-up to Definition 2 is hard to follow. It might be helpful to explicitly state that the LDS score considers the sum of attributions as an additive proxy for $\mathcal{F}$, as it's explained in the original datamodels work.

- In Section 2.3, the authors write that retraining-based methods offer better efficacy than gradient-based methods. Is there existing work that makes this point, or or is this an allusion to results showing that D-TRAK/TRAK improve when using retraining runs? I'm not clear on whether this point is generally true or only true for TRAK, so it could be helpful to clarify.

- Many of the results reflect that the LDS score depends strongly on the number of time steps used when approximating the expectation $\mathbb{E}_t$. It therefore seems like the results entangle two separate concerns: the intrinsic correctness of the attribution method, and the efficiency of estimation. Both matter, but interpreting the results is more difficult because we don't know if either method has converged to its theoretical value due to imprecision in the expectation. I wonder if this subject deserves more discussion in the paper. For example, is there a good reason why D-TRAK should be easier to estimate?

- Can the authors explain the consistent drops in LDS scores when shifting from validation examples to generations?

- Related to my main concern described above, do the authors have any other ideas for why their alternative $\mathcal{L}$ formulations make sense? The point in eq. 8 doesn't tell us much. One intuition I have is that it's fundamentally difficult to predict the exact noise value, because there are many "denoising paths" to recover the original image when large noise is added. $\mathcal{L}_{simple}$ therefore seems like a questionable choice to determine whether the model works well for a given image. I'm not sure the new proposals are more sensible, but perhaps they get at, in a limited sense, whether the predicted noise values are at least Gaussian? Further discussion around why they work seems important for the paper.

- On a broader note, can the authors justify their choice to focus on the LDS metric defined via $\mathcal{L}_{simple}$? It seems convenient and natural in a way, because it's how the model is trained, but I don't see an argument that this faithfully represents the model's tendency to generate particular images. I see that this design choice is motivated by prior work, but it seems like a strange implementation decision because it's disconnected from what we really care about, which is whether the model will produce certain generations.

**Questions:**

Several questions are mentioned in the weaknesses section above.

---

> ### Author Response · Authors · 2023-11-17
> **Response to Reviewer 3yAm [1/2]**
>
> Thank you for your supportive review and suggestions. Below we try to offer appropriate explanations for why D-TRAK works and respond to the detailed comments.
>
> ---
>
> ***It would be helpful for the authors to clarify why they can't use ArtBench itself, for example. It seems like the cost of running TRAK/D-TRAK should be manageable, given that retraining is optional in these methods.***
>
> Indeed, running TRAK/D-TRAK is computationally efficient and scalable to larger datasets, given that retraining is optional in these methods. However, to compute LDS scores on a large-scale dataset, such as ArtBench, we must first build the corresponding LDS benchmark, which requires retraining a large number of models and accounts for the majority of the computational cost. We tried our hardest and scaled up to CIFAR10 and ArtBench-5 based on our maximum computation budget.
>
> ---
>
> ***Why do the interpolation in Section 3.2?***
>
> As reported in Table 1, $\\mathcal{L}\_{\\textrm{Square}}$, $\\mathcal{L}\_{\\textrm{Avg}}$, $\\mathcal{L}\_{\\textrm{$2$-norm}}$, and $\\mathcal{L}\_{\\textrm{$1$-norm}}$ consistently outperform $\\mathcal{L}\_{\\textrm{Simple}}$. The counterintuitive results in Table 1 motivate us to take a closer look at the difference between $\\nabla\_{\\theta}\\mathcal{L}\_{\\textrm{Simple}}$ and $\\nabla\_{\\theta}\\mathcal{L}\_{\\textrm{Square}}$. In Eq. (8), we observe that $\\nabla\_{\\theta}\\mathcal{L}\_{\\text{Simple}}$ consists of two terms: $\\mathbb{E}\_{t,\\boldsymbol{\\epsilon}}\\left[2\\cdot\\boldsymbol{\\epsilon}^{\\top}\\nabla\_{\\theta}\\boldsymbol{\\epsilon}\_{\\theta}\\right]$ and $\\mathbb{E}\_{t,\\boldsymbol{\\epsilon}}\\left[2\\cdot\\boldsymbol{\\epsilon}\_{\\theta}^{\\top}\\nabla\_{\\theta}\\boldsymbol{\\epsilon}\_{\\theta}\\right]$, where the latter term is exactly $\\nabla\_{\\theta}\\mathcal{L}\_{\\text{Square}}$. Thus, we propose to interpolate between these two terms to see if there is a combination ratio that leads to a higher LDS score.
>
> Surprisingly, as shown in Figure 1, the most theoretically-justified $\\nabla\_{\\theta}\\mathcal{L}\_{\\textrm{Simple}}$ (i.e., $\\eta=0.5$) consistently has the lowest LDS scores. It seems that the poor performance of $\\mathcal{L}\_{\\textrm{Simple}}$ is a result of the cancel-out effect of the above two terms, which might be related to the gradient obfuscation/saturation problem (as discussed in Section 5).
>
> ---
>
> ***The role of non-convexity.***
>
> Non-convexity is always a major factor leading to inferior performance of theoretically justified (in convex settings) attribution methods such as TRAK. However, non-convexity may not explain why D-TRAK empirically outperforms TRAK, because there is no evidence that D-TRAK should be superior in non-convex settings.
>
> In addition to non-convexity, we notice another factor influencing TRAK performance, which may be associated with gradient obfuscation or saturation. Specifically, as seen in Figure 2, TRAK achieves its highest LDS score on *intermediate* checkpoints, whereas D-TRAK achieves its highest LDS score on *final* checkpoints. This indicates that the gradients of $\\nabla\_{\\theta}\\boldsymbol{\\mathcal{L}}$ used in TRAK tend to saturate as the model converges, which may not be a good design choice for attributing the training objective $\\boldsymbol{\\mathcal{L}}$ itself, as discussed in Section 5.
>
> Given these observations, it is hypothesized that D-TRAK benefits from using loss functions that are not identical but are connected to the training objective. This would prevent gradient obfuscation or saturation while retaining sufficient information for attribution.
>
> ---
>
> ***The role of Shapley values.***
>
> Thank you for pointing out, we have corrected the reference to Ghorbani & Zou [1] in the revision. Regarding the use of Shapley values in evaluation, for both feature attribution and data attribution, the exact Shapley value should be able to be viewed as the ground truth to evaluate different attribution methods that aim to efficiently approximate the exact Shapley values.
>
> ---
>
> ***The lead-up to Definition 2 is hard to follow.***
>
> In the revision, we explicitly state that the LDS score considers the sum of attributions as an additive proxy for $\\boldsymbol{\\mathcal{F}}$ following your suggestion.

---

> ### Author Response · Authors · 2023-11-17
> **Response to Reviewer 3yAm [2/2]**
>
> ***Retraining-based methods offer better efficacy than gradient-based methods.***
>
> This claim follows the TRAK paper [2]. In terms of optimizing LDS scores, retraining-based methods like the datamodels approach [3] accurately predict the model output on a target test input by retraining at most 1.5 million models on subsets of CIFAR-10. This approach also outperforms various methods in terms of predicting data counterfactuals. In Appendix A.1 of the TRAK paper [2], the authors show that datamodels (300K models) are the best when estimating prediction brittleness. Thus, with unlimited computation, retraining-based methods offer better efficacy than gradient-based methods by simply retraining many models to obtain accurate estimation.
>
> ---
>
> ***Is there a good reason why D-TRAK should be easier to estimate than TRAK?***
>
> We deduce that this is because TRAK suffers from gradient obfuscation or saturation as we mentioned above, and therefore TRAK needs to aggregate information from more timesteps to provide effective attribution. In contrast, D-TRAK can obtain sufficient information from much less timesteps.
>
> ---
>
> ***Can the authors explain the consistent drops in LDS scores when shifting from validation examples to generations?***
>
> This is because training examples and validation examples are both sampled from the true data distribution, whereas generated examples are sampled from the model distribution. In practice, there is always a gap between the true data distribution and the learned model distribution, so there are consistent drops in LDS scores when attributing generated data to training data, compared to attributing validation data.
>
> ---
>
> ***Do the authors have any other ideas for why their alternative $\\mathcal{L}$ formulations make sense?***
>
> As reported in Table 1, $\\mathcal{L}\_{\\textrm{Square}}$, $\\mathcal{L}\_{\\textrm{Avg}}$, $\\mathcal{L}\_{\\textrm{$2$-norm}}$, and $\\mathcal{L}\_{\\textrm{$1$-norm}}$ consistently outperform $\\mathcal{L}\_{\\textrm{Simple}}$, while $\\mathcal{L}\_{\\textrm{$\\infty$-norm}}$ has worse performance. We deduce that this is because $\\mathcal{L}\_{\\textrm{Square}}$, $\\mathcal{L}\_{\\textrm{Avg}}$, $\\mathcal{L}\_{\\textrm{$2$-norm}}$, and $\\mathcal{L}\_{\\textrm{$1$-norm}}$ can better retain the information from $\\nabla\_{\\theta}\\boldsymbol{\\epsilon}\_{\\theta}$, while $\\mathcal{L}\_{\\textrm{Simple}}=\\mathbb{E}\_{t,\\boldsymbol{\\epsilon}}[2\\cdot(\\boldsymbol{\\epsilon}\_{\\theta}-\\boldsymbol{\\epsilon})\^{\\top}\\nabla\_{\\theta}\\boldsymbol{\\epsilon}\_{\\theta}]$ becomes ill-behaved when model converges (i.e., $\\boldsymbol{\\epsilon}\_{\\theta}\\approx \\boldsymbol{\\epsilon}$ and thus $\\boldsymbol{\\epsilon}\_{\\theta}-\\boldsymbol{\\epsilon}$ becomes a noisy error term). As to $\\mathcal{L}\_{\\textrm{$\\infty$-norm}}$, it has worse performance since it discards too much information and only retains the maximum value.
>
> ---
>
> ***Can the authors justify their choice to focus on the LDS metric defined via $\\mathcal{L}\_{\\textrm{Simple}}$?***
>
> Indeed, the design choice of defining LDS via $\\mathcal{L}\_{\\textrm{Simple}}$ follows prior work [4], and it seems that $\\mathcal{L}\_{\\textrm{ELBO}}$ may be a more reasonable choice since it provides a lower bound on log-likelihood. Nevertheless, recent work [5] shows that $\\mathcal{L}\_{\\textrm{Simple}}$ can be regarded as a weighted integral of ELBOs over different noise levels, and can be equivalent to ELBO under monotonic conditions.
>
> Besides, in Section 4.4, we empirically perform counterfactual evaluation by measuring the $\\ell_{2}$-distance and CLIP similarity between the images generated before/after the exclusion of top-1000 positive influencers identified by D-TRAK and TRAK. Both the quantitative (Figures 3 and 11) and visualized (Figure 4) results show that D-TRAK is better than TRAK, which is consistent with the trend concluded from the LDS metric.
>
> ---
>
> ***References:*** \
> [1] Data Shapley: Equitable Valuation of Data for Machine Learning. ICML 2019 \
> [2] Trak: Attributing Model Behavior at Scale. ICML 2023 \
> [3] Datamodels: Predicting Predictions from Training Data. ICML 2022 \
> [4] The Journey, Not the Destination: How Data Guides Diffusion Models. ICML Workshop 2023 \
> [5] Understanding Diffusion Objectives as the ELBO with Simple Data Augmentation. NeurIPS 2023

---

> ### Comment · Reviewer_3yAm · 2023-11-20
> **Response**
>
> Thanks to the authors for their response. The clarifications are helpful, but the degree of insight on 1) why these alternative versions of TRAK are easier to estimate and 2) why they may be intrinsically more useful (i.e., even if all versions were perfectly estimated) remains somewhat unsatisfactory. I wonder if the authors could do a better job with these points given more time to revise the work, and perhaps design new experiments.
>
> Furthermore, regarding the last point about the choice of evaluation via LDS with ${\mathcal{L}}_{\text{simple}}$, my question was less about L_simple vs. L_ELBO - I was curious whether either have a clear relationship with the likelihood of a generation under the model's probability distribution. It seems like that's what we should try to assess with data attribution methods for diffusion models: for any generation, we care which training examples made them more likely. The current formulation seems convenient, and I see that it follows from prior work, but this paper doesn't seem to argue that it represents what we really care about.
>
> Overall, I'm not ready to raise my score. In terms of soundness and completeness I'm tempted to lower my score to marginally below acceptance, but I recognize that this is a timely topic that would be of interest to ICLR attendees.

---

> > ### Author Response · Authors · 2023-11-21
> > **Thank you for your timely feedback and suggestions**
> >
> > Thank you for your timely feedback and suggestions. Though we have made every effort to provide an understanding of D-TRAK through empirical ablation studies, we admit that there is still room for further improvement, both empirically and theoretically. We will do our best to design new experiments to gain a better understanding of how D-TRAK works.
> >
> > ---
> >
> > ***The relationship between $\\mathcal{L}\_{\\textrm{Simple}}$ and the likelihood of a image***
> >
> > The reconstruction loss $\\mathcal{L}\_{\\textrm{Simple}}$ on a particular image does faithfully represent the model's tendency to generate the image, i.e. the likelihood for the model to generate the image. This is because $\\mathcal{L}\_{\\textrm{Simple}}$ on a particular image is proportional to the evidence lower bound (ELBO) of the likelihood for the model to generate the image [1, 2, 3].
> >
> > Please note that for DDPMs, we *cannot* compute the exact likelihood or the gradients of likelihood [1, 2]. Therefore, a more tractable proxy for the likelihood is the $\\mathcal{L}\_{\\textrm{Simple}}$ that is used to train the diffusion model given their relationship mentioned above.
> >
> > ---
> >
> > ***References:*** \
> > [1] Deep unsupervised learning using nonequilibrium thermodynamics. ICML 2015 \
> > [2] Denoising Diffusion Probabilistic Models. NeurIPS 2020 \
> > [3] Understanding Diffusion Objectives as the ELBO with Simple Data Augmentation. NeurIPS 2023

---

### Official Review · Reviewer_nsDq · 2023-11-06

**Soundness:** 3 good
**Presentation:** 2 fair
**Contribution:** 2 fair
**Rating:** 6
**Confidence:** 4

**Summary:**

This paper tackles the problem of attributing images generated by diffusion models back to the training data of these models. They make simple modifications to existing methods [1, 2] and report significantly better results on a standard evaluation metric (LDS) for this task. The authors do not provide a justification for the discrepancy in empirical results between their method and [1], and call for further exploration of the design choices within data attribution methods for diffusion models.


[1] Sung Min Park, Kristian Georgiev, Andrew Ilyas, Guillaume Leclerc, and Aleksander Madry. Trak:
Attributing model behavior at scale. In International Conference on Machine Learning (ICML),
2023.

[2] Kristian Georgiev, Joshua Vendrow, Hadi Salman, Sung Min Park, and Aleksander Madry. The
journey, not the destination: How data guides diffusion models. In Workshop on Challenges in
Deployable Generative AI at International Conference on Machine Learning (ICML), 2023.

---

Score raised from 5 to 6 after authors' response.

**Strengths:**

- The authors thoroughly test their proposed method on a number of datasets
- The authors present strong empirical results across a variety of settings

**Weaknesses:**

- Out of the listed baselines, to the best of my knowledge only Journey TRAK [1] has been explicitly used for diffusion models in previous work. As the authors note, Journey TRAK is not meant to be used to attribute the *final* image $x$ (i.e., the entire sampling trajectory). Rather, it is meant to attribute noisy images $x_t$ (i.e., specific denoising steps along the sampling trajectory). Thus, the direct comparison with Journey TRAK in the evaluation section is not on equal grounds.

- For the counterfactual experiments, I would have liked to see a comparison against Journey TRAK [1] used at a particular step of the the sampling trajectory. In particular, [1, Figure 2] shows a much larger effect of removing high-scoring images according to Journey TRAK, in comparison with CLIP cosine similarity.

- Given that the proposed method is only a minor modification of existing methods [1, 2], I would have appreciated a more thorough attempt at explaining/justifying the changes proposed by the authors.

[1] Kristian Georgiev, Joshua Vendrow, Hadi Salman, Sung Min Park, and Aleksander Madry. The
journey, not the destination: How data guides diffusion models. In Workshop on Challenges in
Deployable Generative AI at International Conference on Machine Learning (ICML), 2023.

[2] Sung Min Park, Kristian Georgiev, Andrew Ilyas, Guillaume Leclerc, and Aleksander Madry. Trak:
Attributing model behavior at scale. In International Conference on Machine Learning (ICML),
2023.

**Questions:**

- Why is the rank correlation close to $0$ when $\mathcal{L}\_{square}$ is used for both $\phi^s$ and $\mathcal{F}$ (Table 4)?
The authors have observed that $\mathcal{L}\_{square}$ is useful as a model output function when predicting $\mathcal{L}\_{simple}$ or $\mathcal{L}\_{ELBO}$. It is particularly odd then that it does not do a good job at predicting *itself*. Is there any particular reason why that might be the case?

- Why set $\mathcal{Q}\equiv \frac{\partial\mathcal{L}}{\partial\mathcal{F}}$ to be the identity matrix even when $\mathcal{F}\neq \mathcal{L}$?

---

> ### Author Response · Authors · 2023-11-17
> **Response to Reviewer nsDq [1/2]**
>
> Thank you for your valuable review and suggestions, we have uploaded a paper revision including additional experiment results. Below we respond to the comments in Weaknesses (***W***) and Questions (***Q***).
>
> ---
>
> ***W1: The direct comparison with Journey TRAK in the evaluation section is not on equal grounds.***
>
> As you mentioned, only Journey TRAK (J-TRAK) [1] has been explicitly used for diffusion models in previous work, but still not meant to be used to attribute the final image. This means that *there is no off-the-shelf baseline for D-TRAK*, so we have to actively adapt previous methods (including TRAK and J-TRAK), in order to construct baselines on attributing final images generated by diffusion models.
>
> Although we have made every effort to adapt previous methods as described in Appendix A.3, it is possible that stronger baselines could still be developed. We will make more rigorous clarification on the selection of baselines in the final reversion.
>
> ---
>
> ***W2: A comparison against Journey TRAK on counterfactual experiments.***
>
> Following your suggestions, we perform counterfactual experiments to compare D-TRAK and J-TRAK at specific steps of the sampling trajectory (please see Appendix E for details). As shown in Figures 21, 22, 23, and 24, J-TRAK and its ensembling version J-TRAK(8) can indeed induce a large effect on the generation results via removing high-scoring training images specific to a timestep. Nevertheless, our findings indicate that D-TRAK remains more effective at identifying images that have a larger impact on the generation results at a specific timestep (we evaluate at $t=300,400$ following [1]). Even more counterintuitive are these results in light of the fact that D-TRAK was not originally intended to attribute a specific sampling timestep.
>
> ---
>
> ***W3: A more thorough attempt at explaining/justifying the changes.***
>
> Actually, we happened to find that $\\mathcal{L}\_{\\textrm{Square}}$ counterintuitively works much better than $\\mathcal{L}\_{\\textrm{Simple}}$ and $\\mathcal{L}\_{\\textrm{ELBO}}$ on attributing diffusion models, so we also tried other common $\\ell_{p}$-norm and average pooling losses. As reported in Table 1, $\\mathcal{L}\_{\\textrm{Square}}$, $\\mathcal{L}\_{\\textrm{Avg}}$, $\\mathcal{L}\_{\\textrm{$2$-norm}}$, and $\\mathcal{L}\_{\\textrm{$1$-norm}}$ consistently outperform $\\mathcal{L}\_{\\textrm{Simple}}$, while $\\mathcal{L}\_{\\textrm{$\\infty$-norm}}$ has worse performance. We deduce that this is because $\\mathcal{L}\_{\\textrm{Square}}$, $\\mathcal{L}\_{\\textrm{Avg}}$, $\\mathcal{L}\_{\\textrm{$2$-norm}}$, and $\\mathcal{L}\_{\\textrm{$1$-norm}}$ can better retain the information from $\\nabla\_{\\theta}\\boldsymbol{\\epsilon}\_{\\theta}$, while $\\mathcal{L}\_{\\textrm{Simple}}=\\mathbb{E}\_{t,\\boldsymbol{\\epsilon}}[2\\cdot(\\boldsymbol{\\epsilon}\_{\\theta}-\\boldsymbol{\\epsilon})\^{\\top}\\nabla\_{\\theta}\\boldsymbol{\\epsilon}\_{\\theta}]$ becomes ill-behaved when model converges (i.e., $\\boldsymbol{\\epsilon}\_{\\theta}\\approx \\boldsymbol{\\epsilon}$ and thus $\\boldsymbol{\\epsilon}\_{\\theta}-\\boldsymbol{\\epsilon}$ becomes a noisy error term). As to $\\mathcal{L}\_{\\textrm{$\\infty$-norm}}$, it has worse performance since it discards too much information and only retains the maximum value.
>
> Furthermore, non-convexity is always a major factor leading to inferior performance of theoretically justified (in convex settings) attribution methods such as TRAK. However, non-convexity may not explain why D-TRAK empirically outperforms TRAK, because there is no evidence that D-TRAK should be superior in non-convex settings. In addition to non-convexity, we notice another factor influencing TRAK performance, which may be associated with gradient obfuscation or saturation. Specifically, as seen in Figure 2, TRAK achieves its highest LDS score on *intermediate* checkpoints, whereas D-TRAK achieves its highest LDS score on *final* checkpoints. This indicates that the gradients of $\\nabla\_{\\theta}\\boldsymbol{\\mathcal{L}}$ used in TRAK tend to saturate as the model converges, which may not be a good design choice for attributing the training objective $\\boldsymbol{\\mathcal{L}}$ itself, as discussed in Section 5.
>
> Given these observations, it is hypothesized that D-TRAK benefits from using loss functions that are not identical but are connected to the training objective. This would prevent gradient obfuscation or saturation while retaining sufficient information for attribution.

---

> ### Author Response · Authors · 2023-11-17
> **Response to Reviewer nsDq [2/2]**
>
> ***Q1: Why is the rank correlation close to 0 when $\\mathcal{L}\_{\\textrm{Square}}$ is used for both  $\\phi\^{s}$ and $\\boldsymbol{\\mathcal{F}}$ (Table 4)?***
>
> We have discussed this phenomenon in Appendix B (the paragraph of $\\boldsymbol{\\mathcal{F}}$ selection). Specifically, as mentioned in Appendices E.1 and E.6 of the TRAK paper [2], there is a latent assumption of *linearity* on the model output function $\\boldsymbol{\\mathcal{F}}$ in linear datamodeling prediction tasks. If the chosen $\\boldsymbol{\\mathcal{F}}$ is not well approximated by a linear function of training examples, then that *puts an upper bound on the predictive performance of any attribution method.*
>
> Therefore, note that $\\boldsymbol{\\mathcal{F}}=\\mathcal{L}\_{\\textrm{Simple}}$ can be linearly approximated since its value is small when the model converges (the model is trained by $\\boldsymbol{\\mathcal{L}}=\\mathcal{L}\_{\\textrm{Simple}}$). In contrast, the value of $\\boldsymbol{\\mathcal{F}}=\\mathcal{L}\_{\\textrm{Square}}$ remains large and cannot be well approximated by a linear function.
>
> ---
>
> ***Q2: Why set $\\mathcal{Q}$ to be the identity matrix even when $\\boldsymbol{\\mathcal{F}}\\neq\\boldsymbol{\\mathcal{L}}$?***
>
> In our paper, we primarily focus on the settings where  $\\boldsymbol{\\mathcal{F}}=\\boldsymbol{\\mathcal{L}}$ and notice that the weighting term $\\mathcal{Q}$ becomes an identity matrix in TRAK. When we generalize TRAK to D-TRAK, we simply drop the weighting term $\\mathcal{Q}$ and discover that the performance is already satisfactory. As a result, there is *no weighting term $\\mathcal{Q}$ in D-TRAK* when we conduct ablation studies on D-TRAK in Table 4.
>
> ---
>
> ***References:*** \
> [1] The Journey, Not the Destination: How Data Guides Diffusion Models. ICML Workshop 2023 \
> [2] Trak: Attributing Model Behavior at Scale. ICML 2023

---

> > ### Comment · Reviewer_nsDq · 2023-11-19
> > **Response**
> >
> > I am satisfied with the authors' response and I raise my score.
> >
> > ## Response to W2
> > I appreciate the authors adding these experiments, I find the results very interesting. in particular:
> > 1. It is interesting to see that D-TRAK remains effective when used at a particular timestep. To me, this further corroborates the authors' findings.
> >
> > 2. Regarding:
> > > Take the 4th row in both Figures 22 and 24 as an example, for the timestep 400, J-TRAK (8) leads the diffusion model to generate a man with the white beard. However, for the timestep 300, it is a man with brown bread and an extra hat. In contrast, D-TRAK removes the same set of influential training examples for different timesteps and thus produces a consistent generation: a still-life painting for both timestep 400 and 300. The above phenomenon again highlights the differences between Journey TRAK and D-TRAK.
> >
> >      While this consistency of D-TRAK across timesteps is somewhat expected given the choice of $\mathcal{L}$, it is intriguing to see it holding up in practice.
> >
> > A few follow-up questions:
> > 1. What is the compute budget used for D-TRAK in Appendix E? Specifically, how many noise resampling (and how many models) is it averaged over?
> > 2. Regarding D-TRAK producing the same scores across timesteps: does that hold throughout the entire diffusion trajectory? E.g., are the scores when using timestep $t=0$ consistent with the ones obtained from using $t=1000$? (I understand there may not be enough time during the remainder of the rebuttal period to run additional experiments).

---

> > > ### Author Response · Authors · 2023-11-20
> > > **Thank you for your timely feedback and raising the score**
> > >
> > > Thank you for your timely feedback and raising the score.
> > >
> > > ---
> > >
> > > ***The compute budget used for D-TRAK in Appendix E***
> > >
> > > For both *D-TRAK* and *TRAK*, we apply $1$ noise resampling and $1$ model; for *J-TRAK*, we apply $10$ noise resampling and $1$ model; for *J-TRAK (8)*, we apply $10$ noise resampling and $8$ models. Besides, to approximate the gradients, we average over $100$ timesteps for D-TRAK and TRAK to obtain the scores used across all timesteps along the diffusion trajectory, so the total computational budget of D-TRAK/TRAK in Appendix E is (\# of gradient computations):
> > >
> > > $$100 \\textrm{ timesteps} \\times 1 \\textrm{ noise resampling} \\times 1 \\textrm{ model}\\textrm{.}$$
> > >
> > > In contrast, the total computational budgets of J-TRAK and J-TRAK (8) are (\# of gradient computations):
> > >
> > > $$1 \\textrm{ timestep} \\times 10 \\textrm{ noise resampling} \\times 1 \\textrm{ model} \\times \\textrm{\\# of Journey timesteps}\\textrm{;}$$
> > >
> > > $$1 \\textrm{ timestep} \\times 10 \\textrm{ noise resampling} \\times 8 \\textrm{ model} \\times \\textrm{\\# of Journey timesteps}\\textrm{,}$$
> > >
> > > where \# of Journey timesteps refers to how many timesteps are computed by J-TRAK along the diffusion trajectory (e.g., only calculating $t=300$ and $t=400$ corresponds to $\\textrm{\\# of Journey timesteps}=2$).
> > >
> > > ---
> > >
> > > ***Regarding D-TRAK producing the same scores across timesteps: does that hold throughout the entire diffusion trajectory?***
> > >
> > > Yes, in Appendix E, both D-TRAK and TRAK use the same scores across all timesteps, which hold throughout the entire diffusion trajectory. Specifically, we remove the top-1000 influencers identified by D-TRAK/TRAK at *final* images (i.e., $t=0$); then we retrain the model, run the diffusion trajectory and directly record the noisy generated images at the specific timestep of $t=300$ and $t=400$.

---

### Official Review · Reviewer_yBWG · 2023-11-06

**Soundness:** 4 excellent
**Presentation:** 4 excellent
**Contribution:** 2 fair
**Rating:** 6
**Confidence:** 4

**Summary:**

This paper proposes variants of TRAK for diffusion models by calculating the gradients with respect to various different loss functions and discover that alternative loss functions perform better than the original TRAK in terms of LDS, a data attribution metric.

**Strengths:**

The paper is simple and easy to follow. The extensive experiments on different settings provide solid evidence of D-TRAK performing better than TRAK in terms of LDS. Readers can be easily convinced that there is an issue with either TRAK as a data attribution method or LDS as a data attribution metric.

**Weaknesses:**

Despite the solid experiment results, the desiderata of a data attribution paper is different from an adversarial attack paper. For adversarial attacks, the success of an attack is a sufficient contribution. This could not be said for data attribution. Successfully finding techniques to optimize for a data attribution metric is only meaningful **if the technique reveals insight**, because in practice attackers have no control over the data attribution method. Therefore, unlike writing adversarial attack papers, more insight to explain why changing the loss function would lead to better LDS score should be provided. Is the non-convexity at fault? Is LDS not an appropriate metric? Is there something special about diffusion models that TRAK fails to capture? How does different norm losses lead to better or worse LDS score?

The observation of this paper is extremely interesting. Providing some insights about the success of D-TRAK over TRAK would make it publication-worthy.

**Questions:**

See weaknesses.

---

> ### Author Response · Authors · 2023-11-17
> **Response to Reviewer yBWG**
>
> Thank you for your valuable review and suggestions. Below we try to provide some insights about the success of D-TRAK over TRAK, and respond to the comments in Weaknesses.
>
> ---
>
> ***Is LDS not an appropriate metric?***
>
> We deem LDS as an appropriate metric based on two considerations:
>
> - First, both D-TRAK and TRAK satisfy the *additive* assumption of LDS [1,2], so it is reasonable to follow LDS in evaluating (from the aspect of Spearman rank correlation) the counterfactual predictors constructed from D-TRAK and TRAK.
>
> - Second, in Section 4.4, we empirically perform counterfactual evaluation by measuring the $\\ell_{2}$-distance and CLIP similarity between the images generated before/after the exclusion of top-1000 positive influencers identified by D-TRAK and TRAK. Both the quantitative (Figures 3 and 11) and visualized (Figure 4) results show that D-TRAK is better than TRAK, which is consistent with the trend concluded from the LDS metric.
>
> ---
>
> ***Is the non-convexity at fault?***
>
> Partially yes, non-convexity is always a major factor leading to inferior performance of theoretically justified (in convex settings) attribution methods such as TRAK. However, non-convexity may not explain why D-TRAK empirically outperforms TRAK, because there is no evidence that D-TRAK should be superior in non-convex settings.
>
> In addition to non-convexity, we notice another factor influencing TRAK performance, which may be associated with gradient obfuscation or saturation. Specifically, as seen in Figure 2, TRAK achieves its highest LDS score on *intermediate* checkpoints, whereas D-TRAK achieves its highest LDS score on *final* checkpoints. This indicates that the gradients of $\\nabla\_{\\theta}\\boldsymbol{\\mathcal{L}}$ used in TRAK tend to saturate as the model converges, which may not be a good design choice for attributing the training objective $\\boldsymbol{\\mathcal{L}}$ itself, as discussed in Section 5.
>
> Given these observations, it is hypothesized that D-TRAK benefits from using loss functions that are not identical but are connected to the training objective. This would prevent gradient obfuscation or saturation while retaining sufficient information for attribution.
>
> ---
>
> ***Is there something special about diffusion models that TRAK fails to capture?***
>
> As discussed in Section 5, our main insight is that when the model output function of interest $\\boldsymbol{\\mathcal{F}}$ is the same as the training objective $\\boldsymbol{\\mathcal{L}}$, the gradients of $\\nabla\_{\\theta}\\boldsymbol{\\mathcal{L}}$ may not be a good design choice for attributing $\\boldsymbol{\\mathcal{L}}$ itself. Although in this paper we focus on diffusion models, we deduce that *this insight also holds on classification models.* For example, [2] has reported that using $\\nabla\_{\\theta}\\boldsymbol{\\mathcal{L}}$ in TRAK leads to worse performance than using $\\nabla\_{\\theta}\\log(\\exp(\\boldsymbol{\\mathcal{L}}(\\boldsymbol{x};\\theta))-1)$ on classification problems, even if the LDS score is computed by setting $\\boldsymbol{\\mathcal{F}}=\\boldsymbol{\\mathcal{L}}$. Besides, selecting intermediate checkpoints has also been discussed when attributing classifiers [3].
>
> ---
>
> ***How do different norm losses lead to better or worse LDS scores?***
>
> We need to clarify that we did not adversarially search for loss functions to optimize the LDS metric. Actually, we happened to find that $\\mathcal{L}\_{\\textrm{Square}}$ counterintuitively works much better than $\\mathcal{L}\_{\\textrm{Simple}}$ and $\\mathcal{L}\_{\\textrm{ELBO}}$, so we also tried other common $\\ell_{p}$-norm and average pooling losses.
>
> As reported in Table 1, $\\mathcal{L}\_{\\textrm{Square}}$, $\\mathcal{L}\_{\\textrm{Avg}}$, $\\mathcal{L}\_{\\textrm{$2$-norm}}$, and $\\mathcal{L}\_{\\textrm{$1$-norm}}$ consistently outperform $\\mathcal{L}\_{\\textrm{Simple}}$, while $\\mathcal{L}\_{\\textrm{$\\infty$-norm}}$ has worse performance. We deduce that this is because $\\mathcal{L}\_{\\textrm{Square}}$, $\\mathcal{L}\_{\\textrm{Avg}}$, $\\mathcal{L}\_{\\textrm{$2$-norm}}$, and $\\mathcal{L}\_{\\textrm{$1$-norm}}$ can better retain the information from $\\nabla\_{\\theta}\\boldsymbol{\\epsilon}\_{\\theta}$, while $\\mathcal{L}\_{\\textrm{Simple}}=\\mathbb{E}\_{t,\\boldsymbol{\\epsilon}}[2\\cdot(\\boldsymbol{\\epsilon}\_{\\theta}-\\boldsymbol{\\epsilon})\^{\\top}\\nabla\_{\\theta}\\boldsymbol{\\epsilon}\_{\\theta}]$ becomes ill-behaved when model converges (i.e., $\\boldsymbol{\\epsilon}\_{\\theta}\\approx \\boldsymbol{\\epsilon}$). As to $\\mathcal{L}\_{\\textrm{$\\infty$-norm}}$, it has worse performance since it discards too much information and only retains the maximum value.
>
> ---
>
> ***References:*** \
> [1] Datamodels: Predicting Predictions from Training Data. ICML 2022 \
> [2] Trak: Attributing Model Behavior at Scale. ICML 2023 \
> [3] Estimating Training Data Influence by Tracing Gradient Descent. NeurIPS 2020

---

> > ### Comment · Reviewer_yBWG · 2023-11-17
> >
> > These are fair arguments. The key insight of ``the information $\nabla_\theta \epsilon_\theta$ of being better retained in the norm-based losses'' should be highlighted in the paper. Currently the difference between the gradient of different loss terms is only briefly mentioned in Sec 3.2. An additional study should also be provided to isolate whether $\epsilon - \epsilon_\theta$ causing the diminishing of $\nabla_\theta \epsilon_\theta$ signal is the main culprit for worse data attribution.

---

> > > ### Author Response · Authors · 2023-11-18
> > > **Thank you for your timely feedback and suggestions**
> > >
> > > Thank you for your timely feedback and suggestions. We have uploaded a new revision that highlights the insight of ''the information of $\\nabla_\\theta\\boldsymbol{\\epsilon}\_\\theta$ is better retained in the norm-based losses'' in Section 3.2. We will reorganize our paper structure in the final revision and leave more space in the main paper to elaborate on this key insight.
> > >
> > > ---
> > >
> > > ***Whether $\\boldsymbol{\\epsilon}-\\boldsymbol{\\epsilon}\_\\theta$ causing the diminishing of $\\nabla_\\theta\\boldsymbol{\\epsilon}\_\\theta$ signal is the main culprit for worse data attribution?***
> > >
> > > Yes, actually the interpolation experiment (Figure 1) in Section 3.2 is a relevant ablation study to verify this hypothesis. Specifically, $\\nabla\_{\\theta}\\mathcal{L}\_{\\text{Simple}}$ consists of two terms as described in Eq. (8): $\\mathbb{E}\_{t,\\boldsymbol{\\epsilon}}\\left[\\boldsymbol{\\epsilon}^{\\top}\\nabla\_{\\theta}\\boldsymbol{\\epsilon}\_{\\theta}\\right]$ and $\\mathbb{E}\_{t,\\boldsymbol{\\epsilon}}\\left[\\boldsymbol{\\epsilon}\_{\\theta}^{\\top}\\nabla\_{\\theta}\\boldsymbol{\\epsilon}\_{\\theta}\\right]$, where the latter term is exactly $\\nabla\_{\\theta}\\mathcal{L}\_{\\text{Square}}$.
> > > Thus, we propose interpolating between these two terms as $\\phi\^{s}(\\boldsymbol{x})=2\\cdot\\mathbb{E}\_{t,\\boldsymbol{\\epsilon}}[(\\eta\\boldsymbol{\\epsilon}\_{\\theta}-(1-\\eta)\\boldsymbol{\\epsilon})\^{\\top}\\nabla\_{\\theta}\\boldsymbol{\\epsilon}\_{\\theta}]$, in order to see if other combination ratios result in higher or lower LDS scores.
> > >
> > > As shown in Figure 1, $\\nabla\_{\\theta}\\mathcal{L}\_{\\textrm{Simple}}$ corresponds to $\\eta=0.5$ where $\\nabla_\\theta\\boldsymbol{\\epsilon}_\\theta$ is weighted by $\\boldsymbol{\\epsilon}\_\\theta-\\boldsymbol{\\epsilon}$, and *it consistently has the lowest LDS scores.* Furthermore, as the value of $\\eta$ deviates from $0.5$, namely, approaching either $0$ (i.e., $\\boldsymbol{\\epsilon}^{\\top}\\nabla\_{\\theta}\\boldsymbol{\\epsilon}\_{\\theta}$) or $1$ (i.e., $\\boldsymbol{\\epsilon}\_{\\theta}^{\\top}\\nabla\_{\\theta}\\boldsymbol{\\epsilon}\_{\\theta}$), the corresponding LDS scores gradually increase. These findings suggest that the poor performance of $\\mathcal{L}\_{\\textrm{Simple}}$ is (mainly) due to the cancel-out effect induced by $\\boldsymbol{\\epsilon}\_\\theta-\\boldsymbol{\\epsilon}$, which causes the diminishing of the $\\nabla\_\\theta\\boldsymbol{\\epsilon}\_\\theta$ signal and worse data attribution.

---

> > > > ### Comment · Reviewer_yBWG · 2023-11-21
> > > >
> > > > Can the authors please clarify what does ``the cancel-out effect induced by $\epsilon_\theta - \epsilon$'' mean exactly? Based on context I am guessing it is the cancellation between $\epsilon_\theta$ and $\epsilon$ when the model has converged and is capable of denoising. If this is the case, does the term $\eta\epsilon_\theta - (1 - \eta)\epsilon$ actually carry information or does any non-zero random vector suffices the retaining of information from $\nabla_\theta \epsilon_\theta$?

---

> > > > > ### Author Response · Authors · 2023-11-21
> > > > >
> > > > > Yes, when the model has converged, $\\boldsymbol{\\epsilon}\_\\theta$ is capable of denoising (i.e., approximately predicting $\\boldsymbol{\\epsilon}$). As a result, $\\boldsymbol{\\epsilon}\_\\theta-\\boldsymbol{\\epsilon}$ is a noisy learning error with some remaining signal because $\\boldsymbol{\\epsilon}\_\\theta$ does not perfectly predict $\\boldsymbol{\\epsilon}$.
> > > > >
> > > > > Inspired by your question, we further conduct experiments that product $\\nabla\_{\\theta}\\boldsymbol{\\epsilon}\_{\\theta}$ with a non-zero random vector $\\boldsymbol{\\omega}$ as $\\boldsymbol{\\omega}^{\top}\\nabla\_{\\theta}\\boldsymbol{\\epsilon}\_{\\theta}$ in D-TRAK, and we observe that *this leads to an extremely low LDS score.* This additional observation indicates that $\\eta\\boldsymbol{\\epsilon}\_{\\theta}-(1-\\eta)\\boldsymbol{\\epsilon}$ actually carry information.

---

> > > > > > ### Comment · Reviewer_yBWG · 2023-11-22
> > > > > >
> > > > > > The extremely low LDS score of applying a random vector is somewhat shocking since according to Figure 1, a wide range of $\eta$ leads to similarly good LDS score. LDS is quite decent even when $\eta = 0$ which corresponds to $\epsilon^{T}  \nabla_\theta \epsilon_{\theta}$. Isn't $\epsilon$ simply sampled from a normal distribution? What kind of information is the ``$\eta \epsilon_\theta - (1 - \eta) \epsilon$'' term carrying when $\eta = 0$?

---

> > > > > > > ### Author Response · Authors · 2023-11-22
> > > > > > >
> > > > > > > Please note that the input to $\\boldsymbol{\\epsilon}\_\\theta$ is $\\sqrt{\\overline{\\alpha}\_{t}}\\boldsymbol{x}+\\sqrt{1-\\overline{\\alpha}_{t}}\\boldsymbol{\\epsilon}$, and the learning task of diffusion model is to make $\\boldsymbol{\\epsilon}\_\\theta$ predict $\\boldsymbol{\\epsilon}$. Thus, when the model converged, there is $\\boldsymbol{\\epsilon}\_\\theta\\approx \\boldsymbol{\\epsilon}$, and consequently $\\boldsymbol{\\epsilon}\_\\theta\^{\\top}\\nabla\_\\theta\\boldsymbol{\\epsilon}\_\\theta \\approx \\boldsymbol{\\epsilon}\^{\\top}\\nabla\_\\theta\\boldsymbol{\\epsilon}\_\\theta$. This is why in Figure 1 the LDS scores of $\\eta=0$ (i.e., $\\boldsymbol{\\epsilon}\^{\\top}\\nabla\_\\theta\\boldsymbol{\\epsilon}\_\\theta$) and $\\eta=1$ (i.e., $\\boldsymbol{\\epsilon}\_\\theta\^{\\top}\\nabla\_\\theta\\boldsymbol{\\epsilon}\_\\theta$) are comparable.
> > > > > > >
> > > > > > > In contrast, a newly sampled random vector $\\boldsymbol{\\omega}$ is *independent* of both $\\boldsymbol{\\epsilon}$ and $\\boldsymbol{\\epsilon}\_\\theta$, thus $\\boldsymbol{\\epsilon}\^{\\top}\\nabla\_\\theta\\boldsymbol{\\epsilon}\_\\theta$ and $\\boldsymbol{\\omega}\^{\\top}\\nabla\_\\theta\\boldsymbol{\\epsilon}\_\\theta$ are different and lead to different LDS scores.

---

> > > > > > > > ### Comment · Reviewer_yBWG · 2023-11-22
> > > > > > > >
> > > > > > > > I see. The symmetry in Figure 1 makes perfect sense then if we view $\epsilon$ and $\epsilon_\theta$ as almost equivalent (when the model converges).
> > > > > > > >
> > > > > > > > As there are no more concerns, I have increased the score from 5 to 6. The reason for 6 as opposed to 8 is the lack of theoretical support for justifying the D-TRAK function choice. Currently the sole argument is based on cancellation but the cancellation can potentially be mitigated in other ways as well. Nevertheless, this is an interesting observation that the research community should also find ``intriguing’’.

---

> > > > > > > > > ### Author Response · Authors · 2023-11-22
> > > > > > > > > **Thank you for your support and raising the score**
> > > > > > > > >
> > > > > > > > > We greatly appreciate your comments and suggestions, which have been extremely beneficial and inspiring to us. We will polish the paper further and incorporate the rebuttal discussions into the final revision. Thank you!

---

### Author Response · Authors · 2023-11-17
**General Response**

We thank all reviewers for their constructive feedback and we have responded to each reviewer individually.
We have uploaded a paper revision including additional experiment results:

- **Figure 21**: Counterfactual evaluation at timestep 400.
- **Figure 22**: Counterfactual visualization at timestep 400.
- **Figure 23**: Counterfactual evaluation at timestep 300.
- **Figure 24**: Counterfactual visualization at timestep 300.

---

### Meta-Review · Area_Chair_KKrt · 2023-12-15

**Metareview:**

This paper extends a recently proposed data attribution method TRAK to diffusion models. The authors show some particular choices of new losses work better than the original TRAK for diffusion models, and verify the effectiveness of the proposed method empirically.

Reviewers all agree that data attribution for diffusion model is an important area and the empirical studies conducted in this paper is quite solid. However, the reviewers also pointed out that the paper does not give a convincing explanation why the proposed loss works, and this question has not been fully addressed after several rounds of discussions. Given that data attribution for diffusion models is an under-explored topic and could be interested to ICLR audience, we recommend acceptance for this paper.

**Justification For Why Not Higher Score:**

The paper proposed several heuristics without sufficient justification why they work for diffusion models.

**Justification For Why Not Lower Score:**

The topic is interesting and under-explored.

---

### Decision · Program_Chairs · 2024-01-16

Accept (poster)